# Bridging Model-Based Optimization and Generative Modeling via Conservative Fine-Tuning of Diffusion Models

**Masatoshi Uehara** [1*]   **Yulai Zhao** [2*]   **Ehsan Hajiramezanali** [1]   **Gabriele Scalia** [1]
**Gokcen Eraslan** [1] **Avantika Lal** [1] **Sergey Levine**[3†] **Tommaso Biancalani** [1 †]
[1]Genentech    [2]Princeton University    [3]UC Berkeley

## Abstract

AI-driven design problems, such as DNA/protein sequence design, are commonly tackled from two angles: generative modeling, which efficiently captures the feasible design space (e.g., natural images or biological sequences), and model-based optimization, which utilizes reward models for extrapolation. To combine the strengths of both approaches, we adopt a hybrid method that fine-tunes cutting-edge diffusion models by optimizing reward models through RL. Although prior work has explored similar avenues, they primarily focus on scenarios where accurate reward models are accessible. In contrast, we concentrate on an offline setting where a reward model is unknown, and we must learn from static offline datasets, a common scenario in scientific domains. In offline scenarios, existing approaches tend to suffer from overoptimization, as they may be misled by the reward model in out-of-distribution regions. To address this, we introduce a conservative fine-tuning approach, BRAID, by optimizing a conservative reward model, which includes additional penalization outside of offline data distributions. Through empirical and theoretical analysis, we demonstrate the capability of our approach to outperform the best designs in offline data, leveraging the extrapolation capabilities of reward models while avoiding the generation of invalid designs through pretrained diffusion models. The main code is available at `https://github.com/masa-ue/RLfinetuning_Diffusion_Bioseq`.

## 1  Introduction

Computational design involves synthesizing designs that optimize a particular reward function. This approach finds applications in various scientific domains, including DNA/RNA/protein design (Sample et al., 2019; Gosai et al., 2023; Wu et al., 2024). While physical simulations are often used in design problems, lacking extensive knowledge of underlying physical processes necessitates solutions that solely rely on experimental data. In these scenarios, we need an algorithm that synthesizes an improved design by utilizing a dataset of past experiments (i.e., *static offline dataset*). Existing research has addressed computational design from two primary angles. The first angle is generative modeling such as diffusion models (Ho et al., 2020), which aim to directly model the distribution of valid designs by emulating the offline data. This approach allows us to model the space of "valid" designs (e.g., natural images, natural DNA sequences, foldable protein sequences (Avdeyev et al., 2023)). The second angle is offline model-based optimization (MBO), which entails learning the reward model from static offline data and optimizing it with respect to design inputs (Brookes et al., 2019; Trabucco et al., 2021; Angermueller et al., 2019; Linder and Seelig, 2021; Fannjiang and

---

[*]Equal contribution:   `uehara.masatoshi@gene.com`, `yz6292@princeton.edu`
[†]Corresponding authors: `svlevine@eecs.berkeley.edu`,`biancalt@gene.com`

38th Conference on Neural Information Processing Systems (NeurIPS 2024).

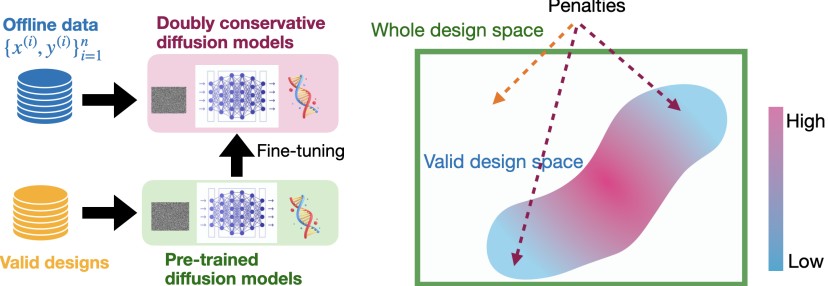

Figure 1: The left figure illustrates our setup with a pre-trained generative model and offline data. On the right, the motivation of the algorithm is depicted. The region surrounded by the green line is the original entire design space, with the colored region indicating the valid design space (e.g., natural images, human-like DNA sequences). The red region denotes areas with more offline data available, while the blue region indicates areas with less data available. We aim to add penalties to the blue regions using conservative reward modeling to prevent overoptimization while imposing a stricter KL penalty on the non-colored regions to prevent the generation of invalid designs.

Listgarten, 2020; Chen et al., 2022). This class of methods potentially enables us to surpass the best design observed in the offline data by harnessing the extrapolative capabilities of reward models.

In our work, we explore how the generative modeling and MBO perspectives could be reconciled, inspired by recent work on RL-based fine-tuning of diffusion models (e.g., Black et al. (2023); Fan et al. (2023)), which aims to finetune diffusion models by optimizing down-stream reward functions. Although these studies do not originally address computational design, we can potentially leverage the strengths of both perspectives. However, these existing studies often focus on scenarios where online reward feedback can be queried or accurate reward functions are available. Such approaches are not well-suited for the typical offline setting, where we lack access to true reward functions and need to rely solely on static offline data (Levine et al., 2020; Kidambi et al., 2020; Yu et al., 2020). In scientific fields, this offline scenario is common due to the high cost of acquiring feedback data. In such contexts, existing works for fine-tuning diffusion models may easily lead to overoptimization, where optimized designs are misled by the trained reward model from the offline data, resulting in out-of-distribution adversarial designs instead of genuinely high-quality designs.

To mitigate overoptimization, we develop a conservative fine-tuning approach for generate models aimed at computational design. Specifically, we consider a critical scenario where we have offline data (with feedback) and a pre-trained diffusion model capable of capturing the space of "valid" designs, and propose a two-stage method (Figure 1). In the initial stage, we train a conservative reward model using offline data, incorporating an uncertainty quantification term that assigns higher penalties to out-of-distribution regions. Subsequently, we finetune pre-trained diffusion models by optimizing the conservative reward model to obtain high-quality designs and prevent the generation of out-of-distribution designs. In the fine-tuning process, we also introduce a KL penalization term to ensure that the generated designs remain within the valid design space.

Our primary contribution lies in the introduction of a novel framework, **BRAID** (douBly conseRvAtive fine-tuning diffusIon moDels). The term "doubly conservative" reflects the incorporation of two types of conservative terms, both in reward modeling and KL penalization. By properly penalizing the fine-tuned diffusion model when it deviates significantly from the offline data distribution, we effectively address overoptimization. Additionally, by framing our fine-tuning procedure within the context of soft-entropy regularized Markov Decision Processes, we offer theoretical justification for the inclusion of these conservative terms in terms of regret. This theoretical result shows that fine-tuned generative models outperform the best designs in the offline data, leveraging the extrapolation capabilities of reward models while avoiding the generation of invalid designs. Furthermore, through empirical evaluations, we showcase the efficacy of our approach across diverse domains, such as DNA/RNA sequences and images.

## 2 Related Works

We summarize related works. For additional works such as fine-tuning on LLMs, refer to Section A.

**Fine-tuning diffusion models via reward functions.** Several previous studies have aimed to improve diffusion models by optimizing reward functions using various methods, including supervised learning (Lee et al., 2023; Wu et al., 2023), RL (Black et al., 2023; Fan et al., 2023) and control-based techniques (Clark et al., 2023; Xu et al., 2023; Prabhudesai et al., 2023). In contrast to our work, their emphasis is not on an offline setting, i.e., their setting assumes online reward feedback is available or accurate reward functions are known. Additionally, while Fan et al. (2023) include the KL term in their algorithms, our innovation lies in integrating conservative reward modeling to mitigate overoptimization and formal statistical guarantees in terms of regret (Theorem 1, 2).

**Conditional diffusion models.** Conditional diffusion models, which learn conditional distributions of designs given the rewards, have been extensively studied (Ho and Salimans, 2022; Dhariwal and Nichol, 2021; Song et al., 2020; Bansal et al., 2023). However, for the purpose of MBO, these approaches require that the offline data has good coverage on values we want to condition on (Brandfonbrener et al., 2022). Compared to conditional diffusion models, our approach aims to obtain designs that can surpass the best design in offline data by leveraging the extrapolation capabilities of reward models. We compare these approaches with our work in Section 7.

**Offline model-based optimization (MBO).** Offline MBO is also known as offline black-box optimization and is closely related to offline contextual bandits and offline RL (Levine et al., 2020). While conservative approaches have been studied there (e.g., Kidambi et al. (2020); Yu et al. (2020) and more in Section A); most of the works are not designed to incorporate a diffusion model, unlike our approach. Hence, it remains unclear how these methods can generate designs that remain within intricate valid design spaces (e.g., generating natural images).

It is worth noting a few exceptions (Yuan et al., 2023; Krishnamoorthy et al., 2023) that attempt to integrate diffusion models into MBO. However, the crucial distinctions lie in the fact that we directly optimize rewards with diffusion models, whereas these prior works focus on using conditional diffusion models. Additionally, we delve into the incorporation of conservative terms, an aspect not explored in their works. We compare these methods with ours empirically in Section 7.

## 3 Preliminaries

We outline our framework for offline model-based optimization with a pre-trained generative model. Subsequently, we highlight the challenges arising from distributional shift. Additionally, we provide an overview of diffusion models, as we will employ them as pre-trained generative models.

### 3.1 Offline Model-Based Optimization with Pre-Trained Generative Model

Our objective is to find a high-quality design within a design space, $\mathcal{X}$. Each design $x \in \mathcal{X}$ is associated with a reward, $r(x)$, where $r : \mathcal{X} \to [0, 1]$ is an unknown reward function. Then, our aim is to find a high-quality generative model $p \in \Delta(\mathcal{X})$, that yields a high $r(x)$. It is formulated as

$$\operatorname{argmax}_{p \in \Delta(\mathcal{X})} \mathbb{E}_{x \sim p}[r(x)]. \tag{1}$$

**Avoiding invalid designs.** In MBO, the design space $\mathcal{X}$ is typically huge. However, in practice, the valid design space denoted by $\mathcal{X}_{\mathrm{pre}}$ is effectively contained within this extensive $\mathcal{X}$ as a potentially lower-dimensional manifold. For instance, in biology, our focus often centers around discovering highly bioactive protein sequences. While the raw search space might encompass $|20|^B$ possibilities (where $B$ is the length), the actual design space corresponding to valid proteins is significantly more constrained. Consequently, our problem can be formulated as:

$$\operatorname*{argmax}_{p \in \Delta(\mathcal{X}_{\mathrm{pre}})} \mathbb{E}_{x \sim p}[r(x)], \left(\text{eqivaletnly, } \operatorname*{argmax}_{p \in \Delta(\mathcal{X})} \mathbb{E}_{x \sim p}[r(x)] - \mathbb{E}_{x \sim p}[\mathrm{I}(x \notin \mathcal{X}_{\mathrm{pre}})]\right). \tag{2}$$

Note supposing that a reward $r(\cdot)$ is 0 outside of $\mathcal{X}_{\mathrm{pre}}$, this is actually still equivalent to (1).

**Offline data with a pre-trained generative model.** Based on the above motivation, we consider scenarios where we have an offline dataset $\mathcal{D}_{\text{off}}$, used for learning the reward function. More specifically, the dataset, $\mathcal{D}_{\text{off}} = \{x^{(j)}, y^{(j)}\}_{j=1}^{n_{\text{off}}}$ contains pairs of designs $x \sim p_{\text{off}}(\cdot)$ and their associated noisy reward feedbacks $y = r(x) + \epsilon$, where $\epsilon$ is noise.

Compared to settings in many existing papers on MBO, we also assume access to a pre-trained generative model (diffusion model) trained on a large dataset comprising valid designs, in addition to the offline data $\mathcal{D}_{\text{off}}$. For example, in biology, this is expected to capture the valid design space $\mathcal{X}_{\text{pre}}$ such as human DNA sequences or physically feasible proteins (Avdeyev et al., 2023; Li et al., 2024; Sarkar et al., 2024; Stark et al., 2024; Campbell et al., 2024). These pre-trained generative models are anticipated to be beneficial for narrowing down the raw search space $\mathcal{X}$ to the design space $\mathcal{X}_{\text{pre}}$. In our work, denoting the distribution induced by the pre-trained model by $p_{\text{pre}}$, we regard the support of $p_{\text{pre}}$ as $\mathcal{X}_{\text{pre}}$.

## 3.2 Challenge: Distributional Shift

To understand our challenges, let's first examine a simple approach for MBO with a pre-trained generative model. For instance, we can adapt methods from Clark et al. (2023); Prabhudesai et al. (2023) to our scenario. This approach involves two steps. In the first step, we perform reward learning: $\hat{r} = \text{argmin}_{\tilde{r} \in \mathcal{F}} \sum_{i=1}^{n_{\text{off}}} \{\tilde{r}(x^{(i)}) - y^{(i)}\}^2$, where $\mathcal{F}$ represents a function class that includes mappings from $\mathcal{X}$ to $[0, 1]$, aiming to capture the true reward function $r(\cdot)$. Then, in the second step, we fine-tune a pre-trained diffusion model to optimize $\hat{r}$.

Despite its simplicity, this approach faces two types of distributional shifts. Firstly, the fine-tuned generative model might produce invalid designs outside of $\mathcal{X}_{\text{pre}}$. As discussed in Section 3.1, we aim to prevent this situation. Secondly, the fine-tuned generative model may over-optimize $\hat{r}$, exploiting uncertain regions of the learned model $\hat{r}$. Indeed, in regions not covered by offline data distribution $p_{\text{off}}$, the learned reward $\hat{r}$ can easily have higher values, while the actual reward values in terms of $r$ might be lower due to the higher uncertainty. We aim to avoid situations where we are misled by out-to-distribution adversarial designs.

## 3.3 Diffusion Models

We present an overview of denoising diffusion probabilistic models (DDPM) (Song et al., 2020; Ho et al., 2020; Sohl-Dickstein et al., 2015). Note while the original diffusion model was initially introduced in Euclidean spaces, it has since been extended to simplex spaces for biological sequences (Avdeyev et al., 2023), which we will use in Section 7. In diffusion models, the goal is to develop a generative model that accurately emulates the data distribution from the dataset. Specifically, denoting the data distribution by $p_{\text{pre}} \in \Delta(\mathcal{X})$, a DDPM aims to approximate using a parametric model structured as $p(x_0; \theta) = \int p(x_{0:T}; \theta) dx_{1:T}$, where $p(x_{0:T}; \theta) = p_{T+1}(x_T; \theta) \prod_{t=T}^{1} p_t(x_{t-1}|x_t; \theta)$. Here, each $p_t$ is considered as a policy, which is a mapping from a design space $\mathcal{X}$ to a distribution over $\mathcal{X}$. By optimizing the variational bound on the negative log-likelihood, we can obtain a set of policies $\{p_t\}_{t=T+1}^{1}$ such that $p(x_0; \theta) \approx p_{\text{pre}}(x_0)$. For simplicity, in this work, assuming that pre-trained diffusion models are accurate, we denote the pre-trained policy as $\{p_t^{\text{pre}}(\cdot|\cdot)\}_{t=T+1}^{1}$, and the generated distribution by the pre-trained diffusion model at $x_0$ by $p_{\text{pre}}$. With slight abuse of notation, we often denote $p_{T+1}^{\text{pre}}(\cdot)$ by $p_{T+1}^{\text{pre}}(\cdot|\cdot)$

# 4 Doubly Conservative Generative Models

We've discussed how naïve approaches for computational design may yield invalid designs or over-optimize reward functions, with both challenges stemming from distributional shift. Our goal in this section is to develop doubly conservative generative models to mitigate this distributional shift.

## 4.1 Avoiding Invalid Designs

To avoid invalid designs, we begin by considering the following generative model:

$$\frac{\exp(\hat{r}(\cdot)/\alpha)p_{\text{pre}}(\cdot)}{\int \exp(\hat{r}(x)/\alpha)p_{\text{pre}}(x)dx} \quad (:= \underset{p \in \Delta(\mathcal{X})}{\text{argmax}} \mathbb{E}_{x \sim p}[\hat{r}(x)] - \alpha\text{KL}(p\|p_{\text{pre}})), \tag{3}$$

where $\text{KL}(p\|p_{\text{pre}}) = \mathbb{E}_{x\sim p}[\log(p(x)/p_{\text{pre}}(x))]$. In this formulation, the generative model is designed as an optimizer of a loss function composed of two parts: the first component encourages designs with high rewards, while the second component acts as a regularizer penalizing the generative model for generating invalid designs. This formulation is inspired by our initial objective in (2), where we substitute an indicator function with $\log(p/p_{\text{pre}})$. This regularizer takes $\infty$ when $p$ is not covered by $p_{\text{pre}}$, and $\alpha$ governs the strength of the regularizer.

## 4.2 Avoiding Overoptimization

Next, we address the issue of overoptimization. This occurs when we are fooled by the learned reward model in uncertain regions. Therefore, a natural approach is to penalize generative models when they produce designs in uncertain regions.

As a first step, let's consider having an uncertainty oracle $\hat{g} : \mathcal{X} \to [0, 1]$, which is a random variable of $\mathcal{D}_{\text{off}}$. This oracle is expected to quantify the uncertainty of the learned reward function $\hat{r}$.

**Assumption 1** (Uncertainty oracle). *With probability $1 - \delta$, we have*

$$\forall x \in \mathcal{X}_{\text{pre}}; |\hat{r}(x) - r(x)| \leq \hat{g}(x) \tag{4}$$

These calibrated oracles are well-established when using a variety of models such as linear models, Gaussian processes, and neural networks. We will provide detailed examples of such calibrated oracles in Section 4.3. Essentially, as long as the reward model is well-specified (i.e., there exists $\tilde{r} \in \mathcal{F}$ such that $\forall x \in \mathcal{X}_{\text{pre}} : \tilde{r}(x) = r(x)$), we can create such a calibrated oracle.

**Doubly Conservative Generative Models.** Utilizing the uncertainty oracle defined in Assumption 1, we present our proposal:

$$\hat{\pi}_\alpha(\cdot) = \frac{\exp\left((\hat{r} - \hat{g})(\cdot)/\alpha\right) p_{\text{pre}}(\cdot)}{\int \exp\left((\hat{r} - \hat{g})(x)/\alpha\right) p_{\text{pre}}(x)dx} \quad (:= \underset{p \in \Delta(\mathcal{X})}{\arg\max} \underbrace{\mathbb{E}_{x\sim p}[(\hat{r} - \hat{g})(x)]}_{\text{Penalized reward}} - \underbrace{\alpha\text{KL}(p\|p_{\text{pre}})}_{\text{KL Penalty}}). \tag{5}$$

Here, to combat overoptimization, we introduce an additional penalty term $\hat{g}(x)$. This penalty term is expected to prevent $\hat{\pi}_\alpha$ from venturing into regions with high uncertainty because it would take a higher value in such regions. We refer to $\hat{\pi}_\alpha$ as a doubly conservative generative model due to the incorporation of two conservative terms.

An attentive reader might question the necessity of simultaneously introducing two conservative terms. Specifically, the first natural question is whether KL penalties, intended to prevent invalid designs, can replace uncertainty-oracle-based penalties. However, this may not hold true because even if we can entirely avoid venturing outside of $\mathcal{X}_{\text{pre}}$ (support of $p_{\text{pre}}$), we may still output designs on uncertain regions not covered by $p_{\text{off}}$. The second question is whether uncertainty-oracle-based penalties can substitute KL penalties. While it is partly true in situations where the support of $p_{\text{off}}$ is contained within that of $p_{\text{pre}}$, uncertainty-oracle-based penalties, lacking leverage on pre-trained generative models, are ineffective in preventing invalid designs. In contrast, KL penalties are considered a more direct approach to stringently avoid invalid designs by leveraging pre-trained generative models.

## 4.3 Examples of Uncertainty Oracles

**Example 1** (Gaussian processes.). *When we use an RKHS as $\mathcal{F}$ (a.k.a. GPs) associated with a kernel $k(\cdot, \cdot) : \mathcal{X} \times \mathcal{X} \to \mathbb{R}$ (Srinivas et al., 2009), a typical construction of $\hat{r}$ and $\hat{g}$ is*

$$\hat{r}(\cdot) = \mathbf{Y}(\mathbf{K} + \lambda I)^{-1}\mathbf{k}(\cdot), \; \hat{g}(\cdot) = c(\delta)\sqrt{\hat{k}(\cdot, \cdot)},$$

*where $c(\delta) \in \mathbb{R}_{>0}, \lambda \in \mathbb{R}_{>0}$, $\mathbf{k}(x) = [k(x^{(1)}, x), \cdots, k(x^{(n_{\text{off}})}, x)]^\top$,*

$\mathbf{Y} = [y^{(1)}, \cdots, y^{(n_{\text{off}})}], \{\mathbf{K}\}_{p,q} = k(x^{(p)}, x^{(q)}), \hat{k}(x, x') = k(x, x') - \mathbf{k}(x)^\top\{\mathbf{K} + \lambda I\}^{-1}\mathbf{k}(x').$

*Note that when using deep neural networks, by considering the last layer as a feature map, we can still create a kernel (Zhang et al., 2022; Qiu et al., 2022).*

**Example 2** (Bootstrap). *When we use neural networks as $\mathcal{F}$, it is common to use a statistical bootstrap method. Note many variants have been proposed (Chua et al., 2018), and its theory has been analyzed (Kveton et al., 2019). Generally, in our context, we generate multiple models $\hat{r}_1, \cdots, \hat{r}_M$ by resampling datasets, and then consider $\arg\min_i \hat{r}_i$ as $\hat{r} - \hat{g}$.*

---

**Algorithm 1 BRAID** (dou**B**ly conse**R**v**A**tive f**I**ne-tuning **D**iffusion models)

---

1: **Require**: Parameter $\alpha \in \mathbb{R}^+$, a set of policy classes $\{\Pi_t\}$ where $\Pi_t \subset [\mathcal{X} \to \Delta(\mathcal{X})]$, pre-trained diffusion model $\{p_t^{\mathrm{pre}}\}_{t=T+1}^1$.
2: Train a conservative reward model $\hat{r} - \hat{g}$ using an offline data $\mathcal{D}_{\mathrm{off}}$.
3: Update a diffusion model as $\{\hat{p}_t\}_t$ by solving the planning problem:

$$\{\hat{p}_t\}_t = \operatorname*{argmax}_{\{p_t \in \Pi_t\}_{t=T+1}^1} \underbrace{\mathbb{E}_{\{p_t\}}[\hat{r}(x_0) - \hat{g}(x_0)]}_{\text{Penalized reward}} - \alpha \underbrace{\Sigma_{t=T+1}^1 \mathbb{E}_{\{p_t\}}[\mathrm{KL}(p_t(\cdot|x_t)\|p_t^{\mathrm{pre}}(\cdot|x_t))]}_{\text{KL penalty}} \quad (6)$$

where the expectation $\mathbb{E}_{\{p_t\}}[\cdot]$ is taken with respect to $\prod_{t=T+1}^1 p_t(x_{t-1}|x_t)$.
4: **Output**: A policy $\{\hat{p}_t\}_t$

---

## 5 Conservative Fine-tuning of Diffusion Models

In this section, we consider how to sample from a doubly conservative generative model $\hat{\pi}_\alpha$, using diffusion models as pre-trained generative models. Our algorithm is outlined in Algorithm 1. Initially, we learn a penalized reward $\hat{r} - \hat{g}$ from the offline data and set it as a target to prevent overoptimization in (6). Additionally, we integrate a KL regularization term to prevent invalid designs. The parameter $\alpha$ governs the intensity of this regularization term.

Formally, this phase can be conceptualized as a planning problem in soft-entropy-regularized MDPs (Neu et al., 2017; Geist et al., 2019). In this MDP formulation:

- The state space $\mathcal{S}$ and action space $\mathcal{A}$ correspond to the design space $\mathcal{X}$.
- The reward at time $t \in [0, \cdots, T]$ ($\in \mathcal{S} \times \mathcal{A} \to \mathbb{R}$) is provided only at $T$ as $\hat{r} - \hat{g}$.
- The transition dynamics at time $t$ ($\in [\mathcal{S} \times \mathcal{A} \to \Delta(\mathcal{S})]$) is an identity $\delta(s_{t+1} = a_t)$.
- The policy at time $t$ ($\in \mathcal{S} \to \Delta(\mathcal{A})$) corresponds to $p_{T+1-t} : \mathcal{X} \to \Delta(\mathcal{X})$.
- The reference policy at $t$ is a policy in the pre-trained model $p_{T+1-t}^{\mathrm{pre}}$

In these entropy-regularized MDPs, the soft optimal policy corresponds to $\{\hat{p}_t\}$. Importantly, we can analytically derive the fine-tuned distribution in Algorithm 1 and show that it simplifies to a doubly conservative generative model $\hat{\pi}_\alpha$, from which we aim to sample.

**Theorem 1.** *Let $\hat{p}_\alpha(\cdot)$ be an induced distribution from optimal policies $\{\hat{p}_t\}_{t=T+1}^1$ in (6), i.e., $\hat{p}(x_0) = \int \{\prod_{t=T+1}^1 \hat{p}_t(x_{t-1}|x_t)\} dx_{1:T}$ when $\{\Pi_t\}$ is a global policy class ($\Pi_t = \{\mathcal{X} \to \Delta(\mathcal{X})\}$). Then,*

$$\hat{p}_\alpha(x) = \hat{\pi}_\alpha(x).$$

We have deferred to the proof in Section 5.1. While similar results are known in the context of standard entropy regularized RL (Levine, 2018), our theorem is novel because previous studies did not consider pre-trained diffusion models.

**Training algorithms.** Based on Theorem 1, to sample from $\hat{\pi}_\alpha$, what we need to is to solve Equation (6). We can employ any off-the-shelf RL algorithms to solve this planning problem. Given that the transition dynamics are known, and differentiable reward models are constructed in our scenario, a straightforward approach to optimize (6) is to directly optimize differentiable loss functions with respect to parameters of neural networks in policies, as detailed in Appendix B. Indeed, this approach has recently been used in fine-tuning diffusion models (Clark et al., 2023; Prabhudesai et al., 2023), demonstrating its stability and computational efficiency.

**Remark 1** (Novelty of Theorem 1). *A theorem similar to Theorem 1 has been proven for continuous-time diffusion models in Euclidean space (Uehara et al., 2024, Theorem 1). However, the primary distinction lies in the fact that while their findings are restricted to Euclidean space, where diffusion policies take Gaussian polices, our results are not constrained to any specific domain. Hence, for example, our Theorem 1 can handle scenarios where the domain is discrete or lies on the simplex space (Avdeyev et al., 2023) in order to model biological sequences as we do in Section 7.*

## 5.1 Sketch of the Proof of Theorem 1

We explain the sketch of the proof of Theorem 1. The detail is deferred to Theorem C.1.

By induction from $t = 0$ to $t = T + 1$, we can first show

$$\hat{p}_t(x_{t-1}|x_t) = \frac{\exp(v_{t-1}(x_{t-1})/\alpha)p_{t-1}^{\mathrm{pre}}(x_{t-1}|x_t)}{\exp(v_t(x_t)/\alpha)}. \tag{7}$$

Here, $v_t(x_t)$ is a soft optimal value function:

$$\mathbb{E}_{\hat{p}}[\hat{r}(x_0) - \hat{g}(x_0) - \alpha \sum_{k=t}^{1} \mathrm{KL}(p_k(\cdot|x_k)\|p_k^{\mathrm{pre}}(\cdot|x_k))|x_t],$$

which satisfies an equation analogous to the soft Bellman equation: $v_0(x) = \hat{r}(x) - \hat{g}(x)$ and for $t = 1$ to $t = T + 1$,

$$\exp(\frac{v_t(x_t)}{\alpha}) = \int \exp\left(\frac{v_{t-1}(x_{t-1})}{\alpha}\right) p_t^{\mathrm{pre}}(x_{t-1} \mid x_t)dx_{t-1}. \tag{8}$$

Now, we aim to calculate a marginal distribution at $t$ defined by $\hat{p}_t(x_t) = \int\{\prod_{k=t+1}^{t} \hat{p}_k(x_{k-1}|x_k)\}dx_{t+1:T}$. Then, by induction, we can show that

$$\hat{p}_t(x_t) = \exp(v_t(x_t)/\alpha)p_t^{\mathrm{pre}}(x_t)/C \tag{9}$$

where $C$ is a normalizing constant. Indeed, supposing that the above (9) hold at $t$, the equation (9) also holds for $t - 1$ as follows:

$$\int \hat{p}_{t-1}(x_{t-1}|x_t)\hat{p}_t(x_t)dx_t = \exp(v_{t-1}(x_{t-1})/\alpha)p_{t-1}^{\mathrm{pre}}(x_{t-1})/C = \hat{p}_{t-1}(x_{t-1}).$$

Finally, by setting $t = 0$, the statement in Theorem 1 is concluded.

## 6 Regret Guarantee

In this section, our objective is to demonstrate that a policy $\hat{p}_\alpha$ from Algorithm 1 can provably outperform designs in offline data by establishing the regret guarantee.

To assess the performance of our fine-tuned generative model, we introduce the soft-value metric:

$$J_\alpha(p) := \mathbb{E}_{x \sim p}[r(x)] - \alpha\mathrm{KL}(p\|p_{\mathrm{pre}}).$$

This metric comprises two components: the expected reward and a penalty term applied when $p$ produces invalid outputs, as we see in (3). Now, in terms of soft-value $J_\alpha(p)$, our proposal $\hat{p}_\alpha$ offers the following guarantee.

**Theorem 2** (Per-step regret). *Suppose Assumption 1. Then, with probability $1 - \delta$, we have*

$$\forall \pi \in \Delta(\mathcal{X}); \underbrace{J_\alpha(\pi) - J_\alpha(\hat{p}_\alpha)}_{\textit{Per step regret}} \leq 2\sqrt{C_\pi} \times \underbrace{\mathbb{E}_{x \sim p_{\mathrm{off}}}[\hat{g}(x)^2]^{1/2}}_{\textit{Stat}}, \quad C_\pi := \max_{x \in \mathcal{X}_{\mathrm{off}}} \left|\frac{\pi(x)}{p_{\mathrm{off}}(x)}\right|,$$

*where $\mathcal{X}_{\mathrm{off}} = \{x \in \mathcal{X} : p_{\mathrm{off}}(x) > 0\}$. As an immediate corollary,*

$$\mathbb{E}_{x \sim \pi}[r(x)] - \mathbb{E}_{x \sim \hat{p}_\alpha}[r(x)] \leq \alpha\mathrm{KL}(\pi\|p_{\mathrm{pre}}) + 2\sqrt{C_\pi} \times \mathbb{E}_{x \sim p_{\mathrm{off}}}[\hat{g}(x)^2]^{1/2}.$$

In the theorem above, we establish that the per-step regret against a generative model $\pi$ we aim to compete with is small as long as the generative model $\pi$ falls within $\mathcal{X}_{\mathrm{off}}$ and the learned model $\hat{r}$ is calibrated as in Assumption 1. First, the term (Stat) corresponds to the statistical error associated with $\hat{r}$ over the offline data distribution $p_{\mathrm{off}}$. When the model is well-specified, it is upper-bounded by $\sqrt{\bar{d}/n}$, where $\bar{d}$ represents the effective dimension of $\mathcal{F}$, as we will discuss shortly. Secondly, the term $C_\pi$ corresponds to the coverage between a comparator generative model $\pi$ and our generative model $\hat{p}_\alpha$. Hence, it indicates that the performance of our learned $\hat{p}_\alpha$ is at least as good as that of a comparator generative model $\pi$ covered by $p_{\mathrm{off}}$. While this original coverage term $C_\pi$ diverges when $\pi$ goes outside of $\mathcal{X}_{\mathrm{off}}$, we can refine it using the extrapolation capabilities of a function class $\mathcal{F}$, as we will discuss shortly. This refined version ensures that we can achieve high-quality designs that outperform designs in the offline data (i.e., best designs in $\mathcal{X}_{\mathrm{off}}$).

**Example 3.** *We consider a scenario where an RKHS is used for $\mathcal{F}$. Let $\mathcal{F}$ be a model represented by an infinite-dimensional feature $\phi(\cdot)$. Let $\bar{d}$ denote the effective dimension of $\mathcal{F}$ (Valko et al., 2013).*

**Corollary 1** (Informal: Formal characterization is in Section D ). *Assuming that the model is well-specified, with probability $1 - \delta$, we have:*

$$
J_\alpha(\pi) - J_\alpha(\hat{p}_\alpha) \leq \sqrt{\bar{C}_\pi} \times \tilde{O}\left(\sqrt{\frac{\bar{d}^3}{n}}\right), \quad \bar{C}_\pi := \sup_{\kappa : \|\kappa\|_2 = 1} \frac{\kappa^\top \mathbb{E}_{x \sim \pi}[\phi(x)\phi^\top(x)]\kappa}{\kappa^\top \mathbb{E}_{x \sim p_{\text{off}}}[\phi(x)\phi^\top(x)]\kappa}.
$$

*The refinement of the coverage term in $\bar{C}_\pi$ is characterized as the relative condition number between covariance matrices on a generative model $\pi$ and an offline data distribution $p_{\text{off}}$, which is smaller than $C_\pi$. This $\bar{C}_\pi$ could still be finite even if $C_\pi$ is infinite. In this regard, Corollary 1 illustrates that the trained generative model can outperform the best design in the offline data by harnessing the extrapolation capabilities of reward models.*

## 7 Experiments

We perform experiments to evaluate (a) the effectiveness of conservative methods for fine-tuning diffusion models and (b) the comparison of our approach between existing methods for MBO with diffusion models (Krishnamoorthy et al., 2023; Yuan et al., 2023). We will start by outlining the baselines and explaining the experimental setups. Regarding more detailed setups, hyperparameters, architecture of neural networks, and ablation studies, refer to Appendix E.

**Methods to compare.** We compare the following methods in our evaluation. For a fair comparison, we always use the same $\alpha$ in **BRAID** and **STRL**. [3].

- **BRAID (proposed method)**: We consider two approaches: (1) **Bonus**, as in Example 1 by setting a last layer as a feature map and constructing a kernel, (2) **Bootstrap**, as in Example 2.

- **Standard RL (STRL)**: RL-fine-tuning that optimizes the standard $\hat{r}$ without any conservative term, following existing works on fine-tuning (Clark et al., 2023; Prabhudesai et al., 2023).

- **DDOM (Krishnamoorthy et al., 2023)**: We train with weighted classifier-free guidance (Ho and Salimans, 2022) using offline data, conditioning on a class with high $y$ values (top $5\%$) during inference. Note that this method is training from scratch rather than fine-tuning.

- **Offline Guidance (Yuan et al., 2023)**: After training a classifier using offline data, we use guidance (conditional diffusion models) (Dhariwal and Nichol, 2021) on top of pre-trained diffusion models and condition on classes with high $y$ values (top $5\%$) at inference time.

**Evaluation.** We assess the performance of each generative model primarily by visualizing the histogram of true rewards $r(x)$ obtained from the generated samples. For completeness, we include similar histograms for both the pre-trained model (**Pretrained**) and the offline dataset (**Offline**). As for hyperparameter selection, such as determining the strengths of conservative terms/epochs, we adhere to conventional practice in offline RL (e.g., Rigter et al. (2022); Kidambi et al. (2020); Matsushima et al. (2020)) and choose the best one through a limited number of online interactions.

**Remark 2.** *We omit comparisons with pure MBO methods for two reasons: (i) **DDOM**, which we compare against, already demonstrates a good performance across multiple datasets, and (ii) these methods are unable to model complex valid spaces since they do not incorporate state-of-the-art generative models (e.g., stable diffusion), thereby lacking the capability to generate valid designs (e.g., natural images) as we show in Section 7.2.*

### 7.1 Design of Regulatory DNA/RNA Sequences

We examine two publicly available large datasets consisting of enhancers ($n \approx 700k$) (Gosai et al., 2023) and UTRs ($n \approx 300k$) (Sample et al., 2019) with activity levels collected by massively parallel reporter assays (MPRA) (Inoue et al., 2019). These datasets have been extensively used in sequence

---

[3] Regarding the effectiveness of KL-regularization, it has been discussed in Fan et al. (2023); Uehara et al. (2024). Hence, in our work, we focus on the effectiveness of conservatism in reward modeling.

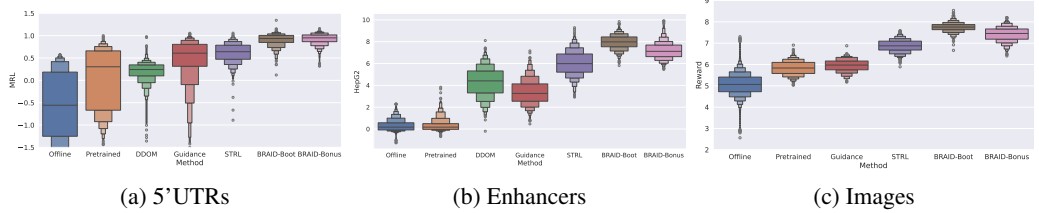

(a) 5'UTRs        (b) Enhancers        (c) Images

Figure 2: Barplots of the rewards $r(x)$ for samples generated by each algorithm. It reveals that proposals consistently outperform baselines.

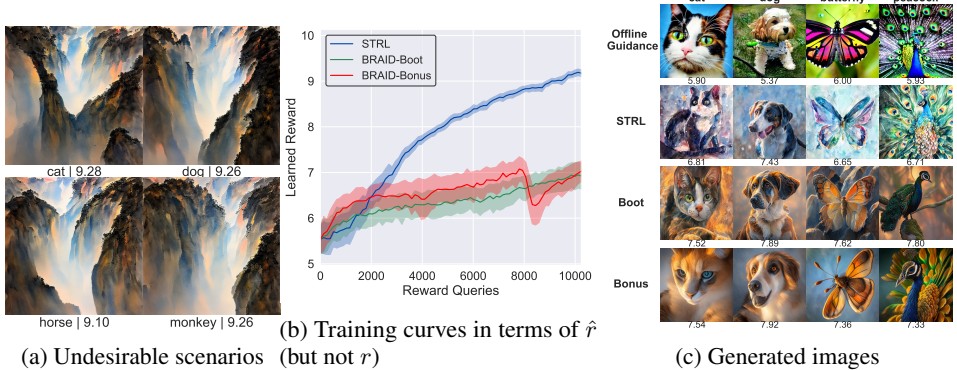

(a) Undesirable scenarios    (b) Training curves in terms of $\hat{r}$ (but not $r$)    (c) Generated images

Figure 3: Results on Image Generation

optimization for DNA and RNA engineering, particularly for the advancement of cell and RNA therapy (Castillo-Hair and Seelig, 2021; Ghari et al., 2023; Lal et al., 2024; Ferreira DaSilva et al., 2024). In the Enhancers dataset, each $x$ is a DNA sequence with a length of 200, while $y \in \mathbb{R}$ is the measured activity in cell lines. For the UTRs dataset, $x$ is a 5'UTR RNA sequence with a length of 50 while $y \in \mathbb{R}$ is the mean ribosomal load (MRL) measured by polysome profiling.

**Setting of oracles and offline data.** We aim to explore a scenario where we have a pre-trained model and an offline dataset. Since the true reward function $r(\cdot)$ is typically unknown, we initially divide the original dataset $\mathcal{D} = \{x^{(i)}, y^{(i)}\}$ randomly into two subsets: $\mathcal{D}_{\text{ora}}$ and $\mathcal{D}'$. Then, from $\mathcal{D}'$, we select datasets below $95\%$ quantiles for enhancers and $60\%$ quantiles for UTRs and define them as offline datasets $\mathcal{D}_{\text{off}}$. Subsequently, we construct an oracle $r(\cdot)$ by training a neural network on $\mathcal{D}_{\text{ora}}$ and use it for testing purposes. Here, we use an Enformer-based model, which is a state-of-the-art model for DNA sequences (Avsec et al., 2021). Regarding pre-trained diffusion models, we use ones customized for sequences over simplex space (Avdeyev et al., 2023). In the subsequent analysis, each algorithm solely has access to the offline data $\mathcal{D}_{\text{off}}$ and a pre-trained diffusion model, but not $r(\cdot)$.

**Results.** The performance results in terms of $r(\cdot)$ are depicted in Fig 2a and b. It is seen that fine-tuned generative models via RL outperform conditioning-based methods: **DDOM** and **Guidance**. This is expected because conditional models themselves are not originally intended to surpass the conditioned value ($\approx$ best value in the offline data). Conversely, fine-tuned generative models via RL are capable of exceeding the best value in offline data by harnessing the extrapolation capabilities of reward modeling, as also theoretically supported in Corollary 1. Secondly, both **BRAID-boot** and **BRAID-bonus** demonstrate superior performance compared to **STRL**. This suggests that conservatism aids in achieving fine-tuned generative models with enhanced rewards while mitigating overoptimization.

### 7.2 Image Generation

We consider the task of generating aesthetically pleasing images, following prior works (Fan et al., 2023; Black et al., 2023). We use Stable Diffusion v1.5 as our pretrained diffusion model, which can generate high-quality images conditioned on prompts such as "cat" and "dog". We use the AVA

dataset (Murray et al., 2012) as our offline data and employ a linear MLP on top of CLIP embeddings to train reward models ($\hat{r}$ and $\hat{r} - \hat{g}$) from offline data for fine-tuning.

**Setting of oracles.** To construct $r(x)$, following existing works, we use the LAION Aesthetic Predictor V2 (Schuhmann, 2022), already pre-trained on a large-scale image dataset. However, this LAION predictor gives high scores even if generated images are almost identical regardless of prompts, as in Figure 3b. These situations are undesirable because it means fine-tuned models are too far away from pre-trained models. Hence, for our evaluation, we define $r(x)$ as follows: (1) asking vision language models (e.g., LLaVA (Liu et al., 2024)) whether images contain objects in the original prompts [4] (e.g., dog, cat), (2) if Yes, outputting the LAION predictor, and (3) if No, assigning 0. This evaluation ensures that high $r(x)$ still indicates capturing the space of the original stable diffusion.

**Results.** We show that our proposed approach outperforms the baselines *in terms of $r(x)$*, as in Fig 2c [5]. We also show the generated images in Figure 3cc. Additionally, we plot the training curve during the fine-tuning process *in terms of the mean of $\hat{r}(x)$* of generated samples in Fig 3cb. The results indicate that in **STRL**, while the learning curve based on the learned reward quickly grows, fine-tuned models no longer necessarily remain within the space of pre-trained models (**STRL** in Fig 2c). In contrast, in our proposal, by carefully regularizing on regions outside of the offline data, we can generate more aesthetically pleasing images than **STRL**, which remain within the space of pre-trained models. For more images/ ablation studies, refer to Appendix E.

## 8 Summary

For the purpose of fine-tuning from offline data, we introduced a conservative fine-tuning approach by optimizing a conservative reward model, which includes additional penalization outside of offline data distributions. Through empirical and theoretical analysis, we demonstrate the capability of our approach to outperform the best designs in offline data, leveraging the extrapolation capabilities of reward models while avoiding the generation of invalid designs through pre-trained diffusion models.

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

# A  Additional Related Works

In this section, we summarize additional related works.

**Conservative approaches in offline RL/offline contextual bandits.** Conservative approaches have been explored in offline RL and contextual bandits. Firstly, in both model-free and model-based RL, one prevalent method involves incorporating an additional penalty on top of the reward functions (Yu et al., 2020; Chang et al., 2021). Secondly, in model-based RL, a common strategy is to train transition dynamics in a conservative manner (Kidambi et al., 2020; Rigter et al., 2022; Uehara and Sun, 2021). Thirdly, in model-free RL, typical approaches include conservative learning of q-functions (Kumar et al., 2020; Xie et al., 2021) or the inclusion of KL penalties against behavioral policies (Wu et al., 2019; Fakoor et al., 2021).

However, these works are not designed to incorporate a diffusion model, unlike our approach. Hence, it remains unclear how their works can generate designs that remain within intricate valid design space, such as high-quality images using stable diffusion.

**Design with generative models.** Many works are focusing on design problems with generative models. However, these works are typically limited to the usage of VAEs. (Notin et al., 2021; Gómez-Bombarelli et al., 2018) Our work is still significantly different because we focus on generative models.

**Fine-tuning in LLMs from human feedbacks.** A closely related area of research involves fine-tuning LLMs through the optimization of reward functions using human feedback (Touvron et al., 2023; Ouyang et al., 2022). Especially, following works such as Zhan et al. (2023), from a theoretical viewpoint, Xiong et al. (2023) explores the effectiveness of pessimism in offline scenarios and its theoretical aspect. However, our theoretical findings are more specifically tailored to diffusion models. Indeed, our main result in Theorem 1 is novel, and our algorithm differs significantly, as fine-tuning methods in offline settings in the literature on LLMs typically rely on policy gradient or PPO, whereas we use more direct backpropagation approaches. Furthermore, the meaning of step size is different as well. Hence, in (Xiong et al., 2023), they do not use soft entropy regularized MDPs.

Typically, in the above works, human feedback is considered to be given in the form of preferences. Similarly, in the context of diffusion models, Yang et al. (2023); Wallace et al. (2023) discusses fine-tuning of diffusion models using preference-based feedback. However, these works focus on online settings but not offline settings.

**Sampling from unnormalized distributions.** In our approach, we use an RL method to sample from our target distribution that is proportional to $\exp((\hat{r} - \hat{g}(x))p_{\mathrm{pre}}(x)$. While MCMC has traditionally been prevalent in sampling from unnormalized Boltzmann distributions $\exp(r(x))$, recent discussions (Zhang and Chen, 2021; Vargas et al., 2023), have explored RL approaches similar to our approach. However, their focus differs from ours, as they do not address the fine-tuning of diffusion models (i.e., no $p_{\mathrm{pre}}$) or sample efficiency in offline settings.

Another relevant literature discusses sampling from unnormalized Boltzmann distributions when pre-trained diffusion models are available (Kong et al., 2024). However, their algorithm closely resembles classifier-based guidance (Dhariwal and Nichol, 2021), rather than a fine-tuning algorithm. Additionally, they do not examine conservatism in an offline setting.

# B  Direct Back Propagation

Our planning algorithm has been summarized in Algorithm 2. Here, we parametrize each policy by neural networks.

For the sake of explanation, we also add typical cases where the domain is Euclidean in Algorithm 3.

---

**Algorithm 2** Direct Back Propagation (General case)

---

1: **Require**: Set a diffusion-model $p(\cdot|x_{t-1};\theta)$, pre-trained model $\{p_{\text{pre}}(\cdot|x_{t-1})\}_{t=T+1}^1$, batch size $m$, a parameter $\alpha \in \mathbb{R}^+$.
2: **Initialize**: $\theta_1 = \theta_{\text{pre}}$
3: **for** $s \in [1, \cdots, S]$ **do**
4:     Set $\theta = \theta_s$.
5:     Collect $m$ samples $\{x_t^{(i)}(\theta)\}_{t=T+1}^0$ from a current diffusion model (i.e., generating by sequentially running polices $\{p_t(\cdot|x_t;\theta)\}_{t=T+1}^1$ from $t = T + 1$ to $t = 1$)
6:     Update $\theta_s$ to $\theta_{s+1}$ by adding the gradient of the following loss $L(\theta)$ with respect to $\theta$ at $\theta_s$:

$$L(\theta) = \frac{1}{m}\sum_{i=1}^m \left[\hat{r}(x_0^{(i)}(\theta)) - \hat{g}(x_0^{(i)}(\theta)) - \alpha\sum_{t=T+1}^1 \text{KL}(p_t(\cdot|x_t;\theta)\|p_{\text{pre}}(\cdot|x_t))\right]. \quad (10)$$

7: **end for**
8: **Output**: Policy $\{p_t(\cdot|\cdot;\theta_S)\}_{t=T+1}^1$

---

---

**Algorithm 3** Direct Back Propagation (in Euclidean space)

---

1: **Require**: Set a diffusion-model $\{\mathcal{N}(\rho(t, x_t;\theta), \sigma_t^2);\theta \in \Theta\}_{t=T+1}^1$, pre-trained model $\{\mathcal{N}(\rho(t, x_t;\theta_{\text{pre}}), \sigma_t^2)\}_{t=T+1}^1$, batch size $m$, a parameter $\alpha \in \mathbb{R}^+$.
2: **Initialize**: $\theta_1 = \theta_{\text{pre}}$
3: **for** $s \in [1, \cdots, S]$ **do**
4:     Set $\theta = \theta_s$.
5:     Collect $m$ samples $\{x_t^{(i)}(\theta)\}_{t=T+1}^0$ from a current diffusion model (i.e., generating by sequentially running polices $\{\mathcal{N}(\rho(t, x_t;\theta), \sigma_t^2)\}_{t=T+1}^1$ from $t = T + 1$ to $t = 1$)
6:     Update $\theta_s$ to $\theta_{s+1}$ by adding the gradient of the following loss $L(\theta)$ with respect to $\theta$ at $\theta_s$:

$$L(\theta) = \frac{1}{m}\sum_{i=1}^m \left[\hat{r}(x_0^{(i)}(\theta)) - \hat{g}(x_0^{(i)}(\theta)) - \alpha\sum_{t=T+1}^1 \frac{\|\rho(x_t^{(i)}(\theta), t;\theta) - \rho(x_t^{(i)}(\theta), t;\theta_{\text{pre}})\|^2}{2\sigma^2(t)}\right].$$
$$(11)$$

7: **end for**
8: **Output**: Policy $\{p_t(\cdot|\cdot;\theta_S)\}_{t=T+1}^1$

---

## C   All Proofs

### C.1   Proof of Theorem 1

Here, we actually prove a stronger statement.

**Theorem 3** (Marginal and Posterior distributions). *Let $\hat{p}_t(x_t)$ and $\hat{p}_t^b(x_t|x_{t-1})$ be marginal distributions at $t$ or posterior distributions of $x_t$ given $x_{t-1}$, respectively, induced by optimal policies $\{\hat{p}_t\}_{T+1}^1$. Then,*

$$\hat{p}_t(x_t) = \exp(v_t(x_t)/\alpha)\hat{p}_t^{\text{pre}}(x_t)/C, \quad \hat{p}_t^b(x_t|x_{t-1}) = \hat{p}_t^{\text{pre}}(x_t|x_{t-1}).$$

**Proof.**    To simplify the notation, we let $f(x) = \hat{r}(x) - \hat{g}(x)$. As a first step, by using induction, we aim to obtain an analytical form of the optimal policy $\{\hat{p}_t(\cdot|x_{t-1})\}$.

First, we define the soft-optimal value function as follows:

$$v_{t-1}(x_{t-1}) = \mathbb{E}_{\{\hat{p}_t\}} \left[f(x) - \alpha\sum_{k=t-1}^1 \text{KL}(\hat{p}_k(\cdot|x_k)\|p_k^{\text{pre}}(\cdot|x_k))|x_{t-1}\right].$$

Then, by induction, we have

$$\hat{p}_t(x_{t-1}|x_t) = \underset{p_t \in \Delta(\mathcal{X})}{\text{argmax}}\, \mathbb{E}_{\{\hat{p}_t\}} \left[v_{t-1}(x_{t-1}) - \alpha\text{KL}(p_t(\cdot|x_t)\|p_t^{\text{pre}}(\cdot|x_t))|x_t\right].$$

With simple algebra, we obtain

$$\hat{p}_t(x_{t-1}|x_t) \propto \exp\left(\frac{v_{t-1}(x_{t-1})}{\alpha}\right) p_t^{\mathrm{pre}}(x_{t-1}|x_t). \tag{12}$$

Here, noting

$$v_t(x_t) = \max_{p_t \in \Delta(\mathcal{X})} \mathbb{E}_{\{\hat{p}_t\}}[v_{t-1}(x_{t-1}) - \alpha\mathrm{KL}(p_t(\cdot|x_t)\|p_t^{\mathrm{pre}}(\cdot|x_t))|x_t],$$

we get the soft Bellman equation:

$$\exp\left(\frac{v_t(x_t)}{\alpha}\right) = \int \exp\left(\frac{v_{t-1}(x_{t-1})}{\alpha}\right) p_t^{\mathrm{pre}}(x_{t-1}|x_t)\mathrm{d}x_{t-1}. \tag{13}$$

Therefore, by plugging (13) into (12), we actually have

$$\hat{p}_t(x_{t-1}|x_t) = \frac{\exp\left(\frac{v_{t-1}(x_{t-1})}{\alpha}\right) p_t^{\mathrm{pre}}(x_{t-1}|x_t)}{\exp\left(\frac{v_t(x_t)}{\alpha}\right)}. \tag{14}$$

Finally, with the above preparation, we calculate the marginal distribution:

$$\hat{p}_t(x_t) := \int \left\{ \prod_{s=T+1}^{t} \hat{p}_s(x_{s-1}|x_s) \right\} dx_{t+1:T+1}.$$

Now, by using induction, we aim to prove

$$\hat{p}_t(x_t) = \exp\left(\frac{v_t(x_t)}{\alpha}\right) p_t^{\mathrm{pre}}(x_t).$$

Indeed, when $t = T + 1$, from (14), this hold as follows:

$$\hat{p}_{T+1}(x_{T+1}) = \frac{1}{C} \exp\left(\frac{v_T(x_{T+1})}{\alpha}\right) p_{T+1}^{\mathrm{pre}}(x_{T+1}).$$

Now, suppose the above holds at $t$. Then, this also holds for $t - 1$:

$$\begin{aligned}
\hat{p}_{t-1}(x_{t-1}) &= \int \hat{p}_t(x_{t-1}|x_t)\hat{p}_t(x_t)\mathrm{d}x_t \\
&= \int \exp\left(\frac{v_{t-1}(x_{t-1})}{\alpha}\right) \{p_t^{\mathrm{pre}}(x_{t-1}|x_t)\}p_t^{\mathrm{pre}}(x_t)\mathrm{d}x_t \quad \text{(Use Equation 14)} \\
&= \exp\left(\frac{v_{t-1}(x_{t-1})}{\alpha}\right) p_{t-1}^{\mathrm{pre}}(x_{t-1}).
\end{aligned}$$

By invoking the above when $t = 0$, the statement is concluded.

### C.2 Proof of Theorem 2

In the following, we condition on the event where

$$\forall x \in \mathcal{X}_{\mathrm{pre}}; |r(x) - \hat{r}(x)| \le g(x).$$

holds.

First, we define

$$\hat{J}_\alpha(\pi) := \mathbb{E}_{x\sim\pi}[\hat{r}(x) - \hat{g}(x)] - \alpha\mathrm{KL}(\pi\|p_{\mathrm{un}}), \quad J_\alpha(\pi) := \mathbb{E}_{x\sim\pi}[r(x)] - \alpha\mathrm{KL}(\pi\|p_{\mathrm{un}}).$$

We note that, $\hat{\pi}_\alpha$ maximizes $\hat{J}_\alpha(\pi)$. Therefore, we have

$$\begin{aligned}
J_\alpha(\pi) - J_\alpha(\hat{\pi}_\alpha) &= J_\alpha(\pi) - \hat{J}_\alpha(\pi) + \hat{J}_\alpha(\pi) - \hat{J}_\alpha(\hat{\pi}_\alpha) + \hat{J}_\alpha(\hat{\pi}_\alpha) - J_\alpha(\hat{\pi}_\alpha) \\
&\le J_\alpha(\pi) - \hat{J}_\alpha(\pi) + \hat{J}_\alpha(\hat{\pi}_\alpha) - J_\alpha(\hat{\pi}_\alpha) \quad\quad \text{(Definition of } \hat{\pi}_\alpha) \\
&\overset{(i)}{\le} J_\alpha(\pi) - \hat{J}_\alpha(\pi). \quad\quad\quad\quad\quad\quad\quad\quad\quad \text{(Pessimism)}
\end{aligned}$$

Here, in the step (i), we use
$$\forall x \in \mathcal{X}_{\mathrm{pre}}; |r(x) - \hat{r}(x)| \le g(x).$$
Then,
$$
\begin{aligned}
J_\alpha(\pi) - J_\alpha(\hat{\pi}_\alpha) \le J_\alpha(\pi) - \hat{J}_\alpha(\pi) &\le 2\mathbb{E}_{x \sim \pi}[\hat{g}(x)] \\
&\le 2\sqrt{\mathbb{E}_{x \sim \pi}[\{\hat{g}(x)\}^2]} \qquad\qquad \text{(Jensen's inequality)} \\
&\le 2\sqrt{\left\|\frac{\pi}{p_{\mathrm{pre}}}\right\|_\infty \mathbb{E}_{x \sim p_{\mathrm{pre}}}[\{\hat{g}(x)\}^2]}. \qquad \text{(Importance sampling)}
\end{aligned}
$$
Hence, the statement is concluded.

## D    Theoretical Guarantees with Gaussian Processes

We explain the theoretical guarantee when using Gaussian processes. In this section, we suppose the model is well-specified.

**Assumption 2.** $y = r(x) + \epsilon$ where $\epsilon \sim \mathcal{N}(0, I)$ where $r$ belongs to an RKHS in $\mathcal{H}_k$.

### D.1    Preparation

We introduce the notation to state our guarantee. For details, see Srinivas et al. (2009, Appendix B), Uehara and Sun (2021, Chapter 6.2), Chang et al. (2021, Chapter C.3).

For simplicity, we first suppose the following.

**Assumption 3.** *The space $\mathcal{X}$ is compact, and $\forall x \in \mathcal{X}; k(x,x) \le 1$.*

We introduce the following definition. Regarding details, refer to Wainwright (2019, Chapter 12).

**Definition 1.** *Let $\mathcal{H}_k$ be the RKHS with the kernel $k(\cdot,\cdot)$. We denote the associated norm and inner product by $\|\cdot\|_k, \langle\cdot,\rangle_k$, respectively. We introduce analogous notations for*
$$\hat{k}(x,x') = k(x,x') - \mathbf{k}(x)^\top \{\mathbf{K} + \lambda I\}^{-1}\mathbf{k}(x').$$
*and denote the norm and inner product by $\|\cdot\|_{\hat{k}}, \langle\cdot,\rangle_{\hat{k}}$.*

Note as explained in Srinivas et al. (2009, Appendix B) and Chang et al. (2021, Chapter C.3), actually, we have $\mathcal{H}_k = \mathcal{H}_{\hat{k}}$.

In the following, We use the feature mapping associated with an RKHS $\mathcal{H}_k$. To define this, from Mercer's theorem, note we can ensure the existence of orthonormal eigenfunctions and eigenvalues $\{\psi_i, \mu_i\}$ such that
$$k(\cdot,\diamond) = \sum_{i=1}^{\infty} \mu_i \psi_i(\cdot)\psi_i(\diamond), \begin{cases} \int \psi_i(x)\psi_i(x)p_{\mathrm{sp}}(x)dx = 1 \\ \int \psi_i(x)\psi_j(x)p_{\mathrm{sp}}(x)dx = 0 (i \ne j) \end{cases}.$$
Then, we define the feature mapping:

**Definition 2** (Feature mapping)**.**
$$\phi(x) := [\sqrt{\mu_1}\psi_1(x), \sqrt{\mu_1}\psi_1(x), \cdots]^\top.$$

Assuming eigenvalues are in non-increasing order, we can also define the effective dimension following Srinivas et al. (2009, Appendix B), Uehara and Sun (2021, Chapter 6.2), Chang et al. (2021, Chapter C.3):

**Definition 3** (Effective dimension)**.**
$$d' = \min_j \left\{ j \in \mathbb{N} : j \ge n \sum_{k=j}^{\infty} \mu_k \right\}.$$

The effective dimension is commonly used and calculated in many kernels (Valko et al., 2013). In finite-dimensional linear kernels $\{x \mapsto a^\top \phi(x) : a \in \mathbb{R}^d\}$ such that $k(x,z) = \phi^\top(x)\phi(z)$, letting $d' := \mathrm{rank}(\mathbb{E}_{x \sim p_{\mathrm{sp}}}[\phi(x)\phi(x)])$, we have
$$d' \le \tilde{d} \le d$$
because there exists $\mu_{\tilde{d}+1} = 0, \mu_{\tilde{d}+2} = 0, \cdots$.

### D.2 Calibrated oracle

Using a result in Srinivas et al. (2009). we show

$$\hat{r}(x) - r(x) \le C(\delta)\sqrt{\hat{k}(x,x)}$$

where

$$C(\delta) = c_1\sqrt{1 + \log^3(n/\delta)\mathcal{I}_n}, \quad \mathcal{I}_n = \log(\det(I + \mathbf{K})).$$

Then, with probability $1 - \delta$, we have

$$
\begin{aligned}
\hat{r}(x) - r(x) &= \langle \hat{r}(\cdot) - r(\cdot), \hat{k}(\cdot, x) \rangle_{\hat{k}} && \text{(Reproducing property)} \\
&\le \|\hat{r}(\cdot) - r(\cdot)\|_{\hat{k}} \times \|\hat{k}(\cdot, x)\|_{\hat{k}} && \text{(CS inequality)} \\
&\le \|\hat{r}(\cdot) - r(\cdot)\|_{\hat{k}} \sqrt{\hat{k}(x,x)} \\
&\le C(\delta)\sqrt{\hat{k}(x,x)}. && \text{(Use Theorem 6 in Srinivas et al. (2009))}
\end{aligned}
$$

### D.3 Regret Guarantee (Proof of Corollary 1)

Recall from the proof of Theorem 3,

$$J_\alpha(\pi) - J_\alpha(\hat{\pi}_\alpha) \le 2\mathbb{E}_{x \sim \pi}[\hat{g}(x)] = 2C(\delta)\mathbb{E}_{x \sim \pi}[\sqrt{\hat{k}(x,x)}].$$

Now, first, to upper-bound $\mathbb{E}_{x \sim \pi}[\sqrt{\hat{k}(x,x)}]$, we borrow Theorem 25 in Chang et al. (2021), which shows

$$\mathbb{E}_{x \sim \pi}[\sqrt{\hat{k}(x,x)}] \le c_1\sqrt{\frac{\tilde{C}_\pi d'\{d' + \log(c_2/\delta)\}}{n}}.$$

where

$$\tilde{C}_\pi := \sup_{\kappa : \|\kappa\|_2 = 1} \frac{\kappa^\top \mathbb{E}_{x \sim \pi}[\phi(x)\phi^\top(x)]\kappa}{\kappa^\top \mathbb{E}_{x \sim p_{\mathrm{sp}}}[\phi(x)\phi^\top(x)]\kappa}$$

Next, in order to upper-bound $C(\delta)$, we borrow Theorem 24 in Chang et al. (2021), which shows

$$\mathcal{I}_n \le c_1\{d' + \log(c_2/\delta)\}d'\log(1+n).$$

The statement in Corollary 1 is immediately concluded.

## E  Additional Details of Experiments

### E.1 DNA/RNA sequences

In this subsection, we add the details of experiments in Section 7.

#### E.1.1 Architecture of Neural Networks

**Diffusion models.** Regarding diffusion models for sequences, we adopt the architecture and algorithm tailored to biological sequences over the simple space (Avdeyev et al., 2023). Its architecture is described in Table 1.

**Oracles.** We use the architecture in Avsec et al. (2021), which is a state-of-the-art model in sequence modeling. We just change the last layer so that it is tailored to a regression problem as in Lal et al. (2024).

#### E.1.2 Hyperparameters

In all experiments, we use A100 GPUs. The important hyperparameters are summarized in the following table (Table 2 on page 22).

Table 1: Basic architecture of networks for diffusion models

| Layer | Input dimension | Output dimension | Explanation |
|-------|-----------------|------------------|-------------|
| 1 | $200 \times 4$ | 256 | Linear + ReLU |
| 2 | 256 | 256 | Conv1D + ReLU |
| ... | ... | ... | ... |
| 10 | 256 | 256 | Conv1D + ReLU |
| 11 | 256 | 256 | ReLU |

Table 2: Important hyperparameters for fine-tuning. For all methods, we use Adam as an optimizer.

| Method | Type | Value |
|--------|------|-------|
| BRAID | Batch size | 128 |
| | KL parameter $\alpha$ | 0.001 |
| | LCB parameter (bonus) $c$ | 0.1 (UTRs), 0.1 (Enhancers) |
| | Number of bootstrap heads | 3 |
| | Sampling to neural SDE | Euler Maruyama |
| | Step size (fine-tuning) | 50 |
| Guidance | Guidance level | 10 |
| | Guidance target | Top 5% |

**Hyperparameter selection.** The process of selecting hyperparameters in offline RL is known to be a challenging task (Rigter et al., 2022; Paine et al., 2020) in general. A common practice in existing literature is determining crucial hyperparameters with a limited number of online interactions. In our case, the key hyperparameters include the strength of the LCB parameter (utilized online in **BRAID-Bonus**) and the termination criteria during training (applied to all fine-tuning algorithms such as **STRL**). To ensure a fair comparison, we operate within a framework where we can utilize $120 \times 20$ online samples. This implies that, for instance, in **STRL** and **BRAID-Bonus**, we conduct an online evaluation using 120 samples for 20 pre-defined epochs. However, in **BRAID-Bonus**, given the additional hyperparameters to be tested (strengths of the bonus term $0.01, 0.1, 1.0$), we use 40 samples for each 20 pre-defined epoch.

### E.1.3 Ablation Studies

We performed ablation studies by varying the strength of the bonus parameter $C$ in Figure 5. We chose the one with the best performance in the main text (See the previous section to see the validity of this procedure).

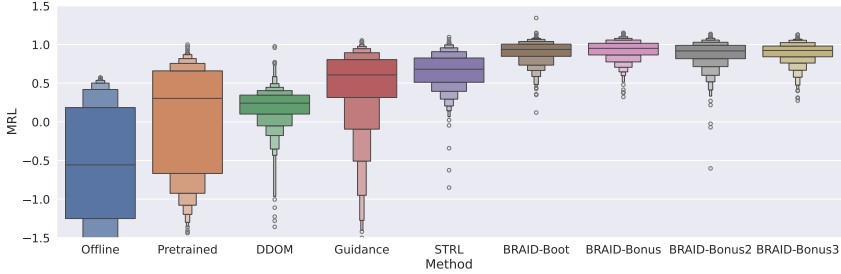

Figure 4: UTRs

### E.2 Images

In this section, we describe the additional experiment regarding image generation in Section 7.2.

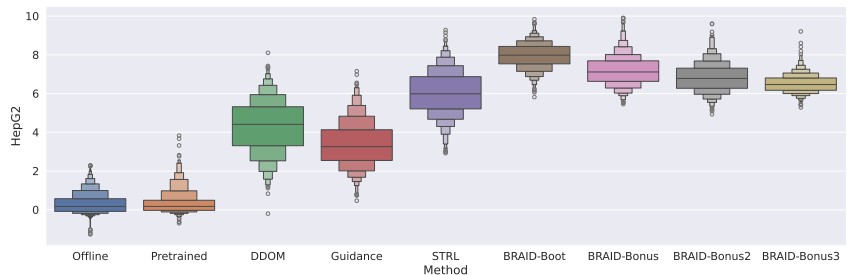

Figure 5: Enhancers

### E.2.1 Description of Offline Data

We utilize images from the AVA dataset (Murray et al., 2012) as samples $x$, containing over $250,000$ image aesthetic evaluations. Rather than using the raw scores directly from the dataset, we derive the labels $y$ by utilizing the pre-trained LAION Aesthetic Predictor V2 Schuhmann (2022) built on top of CLIP embeddings. This choice is made because we employ the LAION Aesthetic Predictor as the ground truth scorer to assess both our methods and baselines. In total, we have curated an offline dataset comprising $255490$ image-score pairs: $\{x^{(i)}, y^{(i)}\}$.

### E.2.2 Architecture of Neural Networks

We adopt the standard StableDiffusion v1.5 (Rombach et al., 2022) as the pre-trained model with the DDIM scheduler (Song et al., 2020). Note this pre-trained model is a conditional diffusion model.

Using the offline dataset $\{(x^{(i)}, y^{(i)})\}$, we train the reward oracle $\hat{r}$ by an MLP on the top of CLIP embeddings. The detailed MLP structure is listed in Table 3. Note that, compared to the true LAION Aesthetic Score Predictor V2 (Schuhmann, 2022), our reward oracle proxy has a simpler structure with fewer hidden dimensions and fewer layers. We aim to impose the hardness of fitting the true reward model, which is typically infeasible in many applications. In such scenarios, a pessimistic reward oracle is especially beneficial to mitigate overoptimization.

Table 3: Architecture of reward oracle for aesthetic scores

| Layer | Input dimension | Output dimension | Explanation |
|---|---|---|---|
| 1 | 768 | 256 | Linear + ReLU |
| 2 | 256 | 64 | Linear + ReLU |
| 3 | 64 | 16 | Linear + ReLU |
| 4 | 16 | 1 | Linear |

### E.2.3 LLM-aided evaluation

As stated in the main text, the original LAION Aesthetic Predictor V2 (Schuhmann, 2022) tends to assign higher scores even to images that disregard the original prompts, which is undesirable. To effectively identify such problematic scenarios, we employ a pre-trained multi-modal language model to verify whether the original prompt is present in the image or not. For each generated image, we provide the following prompt to LLaVA (Liu et al., 2024) along with the image:

```
<image>
USER: Does this image include {prompt}? Answer with Yes or No
ASSISTANT:
```

We evaluated its accuracy and precision with human evaluators by generating images using Stable Diffusion with animal prompts (such as dog or cat). The achieved F1 score was 1.0.

### E.2.4 Hyperparameters

In all image experiments, we use four A100 GPUs for fine-tuning StableDiffusion v1.5 (Rombach et al., 2022). The set of training hyperparameters is listed in Table 4.

Table 4: Important hyperparameters for fine-tuning Aesthetic Scores.

| Method | Parameters | Values |
|---|---|---|
| **BRAID** | Guidance weight | 7.5 |
| | DDIM Steps | 50 |
| | Batch size | 128 |
| | KL parameter $\alpha$ | 1 |
| | LCB bonus parameter $C$ | 0.001 |
| | Number of bootstrap heads | 4 |
| **STRL** | Guidance weight | 7.5 |
| | DDIM Steps | 50 |
| | Batch size | 128 |
| | KL parameter $\alpha$ | 1 |
| **Offline guidance** | Guidance level | 100 |
| | Guidance target | 10 |
| **Optimization** | Optimizer | AdamW |
| | Learning rate | 0.001 |
| | $(\epsilon_1, \epsilon_2)$ | $(0.9, 0.999)$ |
| | Weight decay | 0.1 |
| | Clip grad norm | 5 |
| | Truncated back-propagation step $K$ | $K \sim \text{Uniform}(0, 50)$ |

### E.2.5 Effectiveness of LLaVA-aided evaluation

In our evaluation, we utilize a large multi-modal model like LLaVA. As previously mentioned, relying solely on the raw score fails to detect scenarios where generated images ignore the given prompts.

Table 5 illustrates the outcomes of LLaVA-assisted evaluations for the pre-trained model and four checkpoints of the **STRL** baseline. It is evident that LLaVA successfully identifies all samples generated by the pre-trained model and the first two checkpoints. However, despite seemingly high-reward samples, many samples from checkpoints 3 and 4 do not align correctly with their prompts, resulting in a reduced mean reward. Figure 6 showcases five failure examples from each of checkpoints 3 and 4. Thus, we can validate our quantitative evaluation of reward overoptimization.

Table 5: Statistics of LLaVA-adjusted scores.

| method | mean | min | max | invalid/total samples |
|---|---|---|---|---|
| pre-trained model | 5.789 | 4.666 | 6.990 | 0/400 |
| STRL-ckpt-1 | 6.228 | 4.769 | 7.193 | 0/400 |
| STRL-ckpt-2 | 6.870 | 5.892 | 7.602 | 0/400 |
| STRL-ckpt-3 | 6.484 | 0.0 | 7.944 | 50/400 |
| STRL-ckpt-4 | 0.200 | 0.0 | 7.620 | 389/400 |

### E.2.6 Ablation Studies

**Ablation on BRAID-Bonus hyperparameter** We provide the boxplots for different Bonus hyperparameters in Figure 7, indicating our method's robustness to hyperparameter tuning.

**Additional qualitative results** More image visualizations for **BRAID** and baselines can be found in Figure 8.

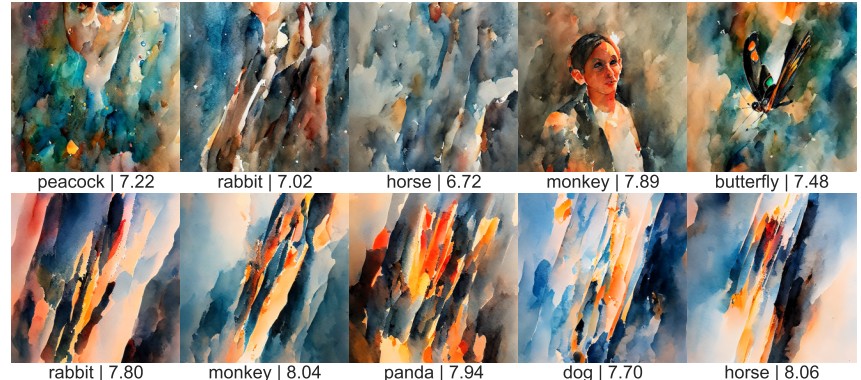

Figure 6: Image-prompt alignment failures detected by LLaVA.

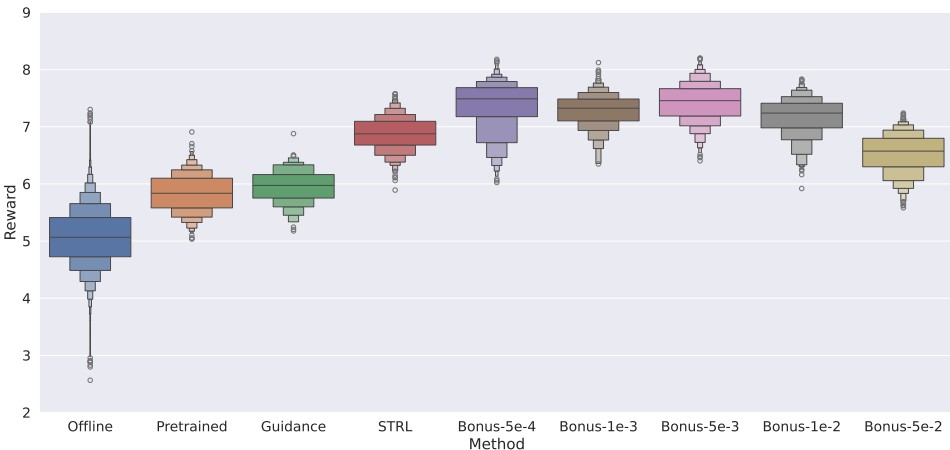

Figure 7: Ablation study for **BRAID-Bonus**. By adjusting the pessimism strength $C_1$ while keeping $\lambda = 0.1$, we show that **BRAID-Bonus** outperforms all baselines for a wide range of hyperparameter selection.

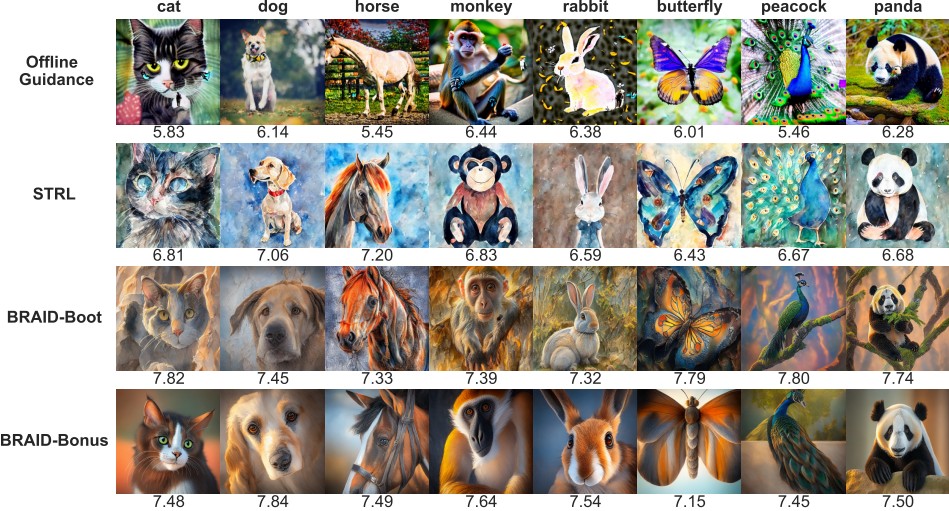

Figure 8: More images generated by **BRAID** and baselines. All algorithms choose the best checkpoint according to our LLaVA-aided evaluation. The visualization demonstrates the benefits of introducing pessimistic terms that can help to achieve high scores while mitigating reward overoptimization.

