# OpenReview forum: "Bridging Model-Based Optimization and Generative Modeling via Conservative Fine-Tuning of Diffusion Models"
_NeurIPS.cc/2024/Conference — NeurIPS 2024 poster_

### Official Review · Reviewer_HnkA · 2024-07-10

**Soundness:** 3
**Presentation:** 4
**Contribution:** 3
**Rating:** 6
**Confidence:** 2

**Summary:**

This paper presents a conservative fine-tuning method called BRAID, which integrates the strengths of diffusion models and model-based optimization (MBO) to improve the performance of pre-trained diffusion models on offline datasets. BRAID optimizes a conservative reward model that includes penalties outside the offline data distribution to prevent overoptimization and generate valid designs. The approach is validated through empirical and theoretical analyses, demonstrating its ability to outperform the best designs in offline data while avoiding the generation of invalid designs. The paper also discusses the method's effectiveness compared to existing conditional diffusion models and traditional MBO techniques, with experiments showcasing its superiority in biological sequence and image generation. The authors acknowledge the limitations of their study, particularly in model selection and hyperparameter tuning, and suggest future research directions.

**Strengths:**

* BRAID incorporates a conservative approach to fine-tuning diffusion models, which includes penalization terms that discourage the model from generating designs outside the distribution of the offline data. This conservative strategy is effective in preventing overoptimization and ensuring the validity of the generated designs.

* The method is supported by both theoretical analysis and empirical results. Theoretically, it provides a regret guarantee, ensuring that the fine-tuned models can outperform the best designs in the offline data. Empirically, it has been validated through experiments across various domains, such as biological sequences and images, demonstrating its ability to generate high-quality designs.

**Weaknesses:**

* Difficulty in tuning hyperparameters without online data interaction.
* Reliance on accurate reward and diffusion models for effective performance.
* Theoretical results depend on certain idealized assumptions that may not hold in all cases.
* Can you compare the methods with other SOTA offline RL methods to illustrate your proposed augmented methods more effective than the SOTA offline RL methods? I think this paper is very relevant to some offline RL methods, such as ReDS[1], A2PR[2], CPED[3], SCQ[4]. It is not required that experimental comparisons must be given, but at least add some discussion with these methods to the paper.

References：

[1] Singh, Anikait, et al. "ReDS: offline reinforcement learning with heteroskedastic datasets via support constraints." Proceedings of the 37th International Conference on Neural Information Processing Systems. 2023.

[2] Liu, Tenglong, et al. "Adaptive Advantage-Guided Policy Regularization for Offline Reinforcement Learning." In International Conference on Machine Learning (ICML). PMLR, 2024.

[3] Zhang, Jing, et al. "Constrained policy optimization with explicit behavior density for offline reinforcement learning." Advances in Neural Information Processing Systems. 2023

[4] Shimizu, Yutaka, et al. "Strategically Conservative Q-Learning." arXiv preprint arXiv:2406.04534 (2024).

**Questions:**

* The methods will need more samples, as shown in the pseudo-code of the algorithms 2 Direct Back Propagation (General case), which may bring more computational burden. Meanwhile, the methods train use two diffusion model to obtain the policy. Can you give me some experiments to show the computational burden?
* Have the authors considered alternative generative models such as GANs or VAEs, and can they provide a comparative analysis of performance and resource usage?
* How to obtain the $\hat{g}$ in the pseudo-code of Algorithm 1 BRAID?
* This methods is related to offline reinforcement learning. So can you give me some experiment comparisions with the SOTA offline RL methods ？

**Limitations:**

* The method requires careful selection of hyperparameters, which can be challenging in a purely offline setting without access to additional online data.
* The pseudo-code of the paper does not illustates the algorithm explicitly. The authors can improve it further.
* This paper lack strong baselines comparisions and enough related works, which is related to some offline reinforcement learning methods. So you can add more related offline reinforcement learning works.

---

> ### Author Rebuttal · Authors · 2024-08-06
>
> Thank you for the detailed and insightful feedback. We have addressed the reviewer's concern by clarifying that (1) our goals differ significantly from those in standard offline RL works,  (2) we have compared our method with recent works that align with our objectives, such as Yuan et al. (2023) and Krishnamoorthy et al. (2023).
>
> **W: Comparison with SOTA offline RL methods, IReDS[1], A2PR[2], CPED[3], SCQ[4].**
>
> Thanks for raising this point. We would like to emphasize that, in general, our originally intended goals are quite different, though we acknowledge that the ideas in the papers you cited are very interesting and potentially helpful in developing new algorithms for our tasks.
>
> 1. **The goals of our paper and most offline RL works are different (as briefly mentioned in Appendix A) because our paper aims not to address standard offline RL tasks but our paper aims to generate high-quality designs in extremely high-dimensional action spaces.** Given the high dimensionality of these spaces (e.g., images, chemicals, biological sequences), our focus is on incorporating SOTA pre-trained generative models (i.e., diffusion models) to constrain the search space to valid domains (natural image space, natural chemical space, natural biological space). While we appreciate the relevance of the works you cited in offline RL, these papers were not designed to address our specific tasks. It is unclear how their methods, which do not include experimental results in our scenarios (such as generating images, biological sequences, or molecules), can be directly applied to our work.
>
> 2. We have compared relevant existing methods in the most pertinent field: Goal-wise, our work aligns more closely with offline (contextual) bandits in extremely high-dimensional action spaces rather than with offline RL. However, the RL aspect appears in our work because diffusion models can be viewed as MDPs, and we have utilized this observation to address our problem as detailed in Section 5. **We have compared several recent baselines, such as Krishnamoorthy et al. (2023) and Yuan et al. (2023), that tackle the same problem as us in Section 7.**
>
> 3. Let me explain whether offline RL works you cited naively address our problem. **While the offline contextual bandit problem is an offline RL problem with horizon 1, it is not obvious that the papers you cited are directly applicable to our task because they do not focus on incorporating expressive pre-trained generative models.** We acknowledge that some insights from these papers could be relevant to our problem. However, translating these insights into our context may not be straightforward and could require more than simply adapting the original algorithms.
>
> - For instance, replacing $\rho$ with diffusion models in Algorithm 1 of ReDS[1] might be valid, but its practical utility is uncertain since the log-likelihood estimation of diffusion models lacks an explicit form and is not differentiable.
> - Similarly, using diffusion models as policies in Algorithm 1 of A2PR[3] might not be practical for our task as well because it is hard to get the explicit form for the log-likelihood.
>
> **W. Difficulty in tuning hyperparameters without online data interaction/Reliance on accurate reward and diffusion models for effective performance**
>
> We agree with the reviewer and have acknowledged our limitations. However, since this limitation is common across many works that use purely offline data (e.g., Rigter et al. (2022); Kidambi et al. (2020)), we have adhered to current conventions in the literature by assuming limited online interaction in actual experimental settings.
>
> **Q: The methods will need more samples, as shown in the pseudo-code of the algorithm 2. Can you give me some experiments to show the computational burden?**
>
> To ensure clarity, let us first distinguish between two types of samples: (a) offline data samples ($x, r(x))$) and (b) artificial samples generated by diffusion models. In the offline setting, sample efficiency in terms of offline data (a) (e.g., lab feedback data) is crucial, so the computational burden (how many samples we use in (b)) is less of a concern as long as it is not excessively high.
>
> **With this in mind, yes, our Algorithm 2 requires artificial samples like (b) above. However, since we are not using offline data, the number of samples used in Algorithm 2 is less concern. We have not discussed the computational complexity as it has already been discussed in  existing works in detail (Black et al. (2023) ,Prabhudesai et al., 2023), and our focus is on sample efficiency.** The actual computational cost varies by domain; for example, image generation might take several hours, while sequence generation might take several minutes. We will address this info in the next version.
>
> **Q: Have the authors considered alternative generative models such as GANs or VAEs, and can they provide a comparative analysis of performance and resource usage?**
>
> This is an interesting point. We will include a more detailed discussion in the Appendix.
>
> - **Reasons for using diffusion models**: In many domains (e.g., images, molecules), diffusion models have demonstrated SOTA performance as generative models, as evidenced by numerous papers (Rombach, Robin, et al; Avdeyev, Pavel, et al). This motivates us to use diffusion models as SOTA generative models to capture the valid space (natural image, chemical, biological spaces).
>
> - **Advantages and disadvantages of GANs and (pure) VAEs:** We acknowledge that GANs and VAEs remain useful as generative models. In particular, fine-tuning to optimize downstream reward functions is computationally faster if we use them. However, as demonstrated in many papers on generative models for images, molecules, the performance as generative models tends to decrease.
>
> **Q: How to obtain the $g$ in the pseudo-code of Algorithm 1 BRAID?**
>
> We have discussed it in Section 4.3. We will add a pointer to make it more accessible to readers.

---

> > ### Comment · Reviewer_HnkA · 2024-08-11
> > **Official Comment by Reviewer HnkA**
> >
> > Thanks for your careful answers and clear explanation. To some extend, my concerns have been addressed. So I expect you to discuss SOTA offline RL as mentioned earlier in your paper and conduct comparison experiments with GANs or VAEs. I would like to keep my score, but it is possible to improve my score.

---

### Official Review · Reviewer_hthV · 2024-07-12

**Soundness:** 4
**Presentation:** 4
**Contribution:** 3
**Rating:** 7
**Confidence:** 3

**Summary:**

The paper tackles the task of black box optimization in an offline setting. Given a pretrained diffusion model, they first train a surrogate model on the offline data and use it to tilt the diffusion model distribution via finetuning it. The authors distinctly focus on an uncertainty quantification based procedure to bias the diffusion model tilting toward regions where the reward is high and the reward uncertainty is low while not tilting toward regions with high uncertainty. Experiments are carried out on a reasonable set of diverse tasks.

The paper introduces a small specific challenge and address it well with a reasonable approach and good motivations. The potential impact of the method may be small but it is neat, educational, and should be useful in many cases. I recommend acceptance.

**Strengths:**

1. Identifying a crucial non-obvious overlooked challenge and specifically pinpointing it. The authors identify that previous work on tilting diffusion models with reward models misses to incorporate tilting less toward regions of the reward function in which it has high uncertainty but high reward. Instead, we should only tilt to the regions that have high reward and high certainty to avoid optimizing toward adversarial examples. (The fact that the finetuned diffusion model will steer away from the pretraining distribution seems like a less relevant insight)
2. The authors identify a relevant overlooked problem in finetuning diffusion models and bring standard techniques from uncertainty quantification into the field to address it in a reasonable fashion. They do not overcomplicate things and their technique could be valuable to several researchers in the area.
3. The authors prove that the training procedure yields the desired distribution. I have no comments regarding the value/insightfulness of the proof. Maybe other reviewers have a stronger opinion about the relevance.
4. The authors evaluate their method on a very diverse set of experiments that includes discrete DNA sequence generation, and image generation. The results are convincing and demonstrate the central empirical claim that out of distribution generation is a problem and is effectively avoided with the proposed conservative reward model fine tuning.
Minor:
1. Interesting snippets of insights. The authors point out interesting relationships and connections along the way which are non-obvious and well placed for putting their motivations into context.
2. Exceptional clarity in writing. The paper lays out the task in its precise specification and covers required concepts and related work in equal clarity.

**Weaknesses:**

1. I would say that the insights in terms of methodological novelty are on the moderate side. The ideas are simple and good, which is appreciated, but the level on which the conceptural changes operate are low level (a tweak to diffusion model tilting) and thus limited in impact. However, it is certainly a good thing to have.
Very hard to address and not a must have for ML conferences:
1. Evaluations are inherently limited in their computational nature and the conclusions that can be drawn for the procedures effectiveness in biological sequence optimization is small. Do you authors disagree with this in any way?

**Questions:**

1. Do I understand correctly that theorem 2 only states that the finetuned diffusion model will incur lower regret than the pretrained diffusion model? Why is that useful and would we not rather be interested in relations between using your additional uncertainty quantification bias for the reward model based finetuning and not using it?
2. It seems to me that reward models are often very poor in scientific applications (more so than the generative models that can be trained on a lot more data). Does this mean that their uncertainty estimates are also bad and your method might not provide any improvements in these cases?

**Limitations:**

The authors point out the key inherent limitation of their work in the fact that the reward models that are often trained on limited data are likely suboptimal instead of just mentioning useless small limitations that are beside the point (which is the more common practice it seems).

---

> ### Author Rebuttal · Authors · 2024-08-06
>
> We appreciate your feedback. We have addressed your concern by explaining a more detailed evaluation plan for biological tasks.
>
> **Weakness: Evaluations are inherently limited in their computational nature and the conclusions that can be drawn for the procedures effectiveness in biological sequence optimization is small. Do you authors disagree with this in any way?**
>
> We thank the reviewer for pointing out that our computational evaluation did not provide enough confidence in our generated biological sequences. While these are complex systems, several of the main factors contributing to the activity of these biological sequences are well-studied and understood. Therefore, there are several analyses we can perform to gain deeper confidence.
>
> * Of the two functions we optimize, enhancer activity in HepG2 is controlled to a large extent by the binding of transcription factors to the DNA sequence, particularly HepG2 specific transcription factors such as HNF4A, HNF4G, HNF1A and HNF1B. 5’UTR activity is, to a large extent, dependent on the sequence of the translation initiation site. Features such as A or G at position −3 relative to AUG and a G at +4 are understood to increase sequence activity, as well as the absence of upstream start codons. These features are understood from many biology studies, including those on the original datasets (Sample et al. 2019, Gosai et al. 2023)
>
> * For the generated HepG2 sequences, we will use the JASPAR database (https://jaspar.elixir.no/) to scan the generated sequences for motifs known to bind to human transcription factors. We expect to see a higher frequency of motifs for activating transcription factors, particularly HepG2- specific ones, and a reduced frequency of motifs known to bind to transcriptional repressors. For the 5’UTR sequences, we will examine the sequences for upstream start codons and also examine base frequencies in the translation initiation sites.
>
> This analysis will clarify whether our model has learned well-established features of biological sequence activity and allow us to draw stronger conclusions regarding the validity of the generated sequences.
>
> **Questions: Do I understand correctly that theorem 2 only states that the finetuned diffusion model will incur lower regret than the pretrained diffusion model? Why is that useful and would we not rather be interested in relations between using your additional uncertainty quantification bias for the reward model based finetuning and not using it?**
>
> Let me clarify as follows.
>
> * Roughly speaking, our theorem (Theorem 2 and Corollary 1) states that our fine-tuned model performs better in terms of $J_{\alpha}$ than the best design in the reward data distribution. Since $J_{\alpha}$  includes a metric related to proximity to pre-trained models, this theorem conveys our key message: 'fine-tuned generative models outperform the best designs in the offline data by leveraging the extrapolation capabilities of reward models while avoiding the generation of invalid designs’.
>
> * The uncertainty quantification term is used to construct $\hat p_{\alpha}$. When evaluating the actual performance in Theorem 2, we have used the true reward $r$.
>
> **Q: It seems to me that reward models are often very poor in scientific applications (more so than the generative models that can be trained on a lot more data). Does this mean that their uncertainty estimates are also bad and your method might not provide any improvements in these cases?**
>
> Yes. While we did not intend to obscure this information, as it is briefly mentioned in the Limitations section, we will make it more explicit.

---

> > ### Comment · Reviewer_hthV · 2024-08-11
> >
> > 1. I appreciate the additional suggestions for experiments and am sure they could be interesting as well. I did not state that "[your] computational evaluation did not provide enough confidence". I think the computational evaluations are well crafted and can provide a fine degree of confidence. I just pointed this out as a minor weakness prepended with "not a must have" (which was maybe overread in the formatting).\
> > Anyways, the additional promised experiments and the qualitative recovery of experimental observations do not change my score. I think it is a good paper and should be accepted. It is outstanding in its clarity of writing (thanks for that effort) but not outstanding overall in my current estimation (I hope that is not rude to say :)).
> >
> > 2. Thanks for taking the time to answer - that was a confused question.
> >
> > 3. "Yes" -> Sad, but expected.

---

### Official Review · Reviewer_dLuW · 2024-07-12

**Soundness:** 2
**Presentation:** 2
**Contribution:** 2
**Rating:** 5
**Confidence:** 4

**Summary:**

This paper proposes a conservative approach for fine-tuning diffusion models with a reward model learned from offline data. Specifically, the ideas are two-fold: The first idea is to replace the reward model with a conservative estimate based on classical generalization bounds. The second idea is to leverage the KL divergence to force the optimized distribution to not deviate too far from the pretrained model. Experiments and theoretical results show the efficacy of the proposed method in fine-tuning diffusion models without over-optimization.

**Strengths:**

1. The proposed method is well-presented, and the motivation behind the algorithm is interesting. The over-optimization problem is indeed critical when fine-tuning diffusion models with learned rewards.

2. Extensive experimental results show the efficacy of the proposed method in improving the reward model while avoiding reward over-optimization.

**Weaknesses:**

1. Leveraging generalization bounds via kernel RKHS and bootstrap is interesting, but I doubt their practicality for real applications. Firstly, the RKHS bound is usually too conservative to be useful, while the computational cost for the bootstrap method is pretty high since one has to train the model from scratch multiple times. As far as I can tell, the reward models used in the experiments are mainly single-layer MLPs, and it is doubtful whether this approach is useful when the reward model needs to be a larger model.

2. Another problem with the conservative estimator of the reward models is that it is unclear whether it is useful given the current experimental results. On one hand, KL regularization is a widely-known technique for preventing over-optimization in diffusion models and is thoroughly studied in existing works, so it is certain that the KL regularization term will help. On the other hand, the proposed algorithm mixes both the conservative reward estimator and the KL regularization term together, making it unclear which part is playing the role in avoiding over-optimization. My guess is that, for the most part, only the KL regularization term is effective in the end.

**Questions:**

STRL methods like AlignProp and DRaFT can work with ODE samplers, which are more commonly used in practice than SDE samplers. However, the method proposed in this work, due to the use of entropy regularization, can only adopt SDE samplers. I wonder if it is possible to design a regularization term for ODE samplers. Could the authors share some insights on this point?

**Limitations:**

See my comments on the weakness above.

---

> ### Author Rebuttal · Authors · 2024-08-06
>
> We appreciate your feedback. We have addressed your concern by explaining (1) the bootstrap/RKHS bonus term's practical usefulness in our scenario and its widespread use in various real applications/papers, and (2) how our experiments are intentionally designed to demonstrate that conservatism (rather than KL) helps mitigate overoptimization. Please let us know if this addresses your concerns, or if there are any other issues we should discuss or address!
>
> **Weakness: RKHS bound is usually too conservative to be useful. I doubt their practicality for real applications.**
>
> We would like to convey that it is still practically useful from several aspects.
>
> 1. **Negative penalty term (Example 1) based on RKHS theory is widely used in the literature on Gaussian processes (GPs)**. For example, as noted in Wikipedia of GPs, many widely used software tools are built around GPs. They have been proven to be effective in real-world applications, e.g., robotics/material science/drug discovery (Deringer et al., Gramacy et al.).
>
> 2. **Our examples with DNA and RNA designs are real applications**, as the tasks, data, and models are all proposed by established scientific journals, as we have cited Sample et al., 19, Inoue et.al, 19. There is more literature on solving this task, e.g., Taskiran et al.
>
> 3. We believe there may be a misunderstanding: **We have combined it with deep learning models.** After extracting features from deep learning models, we used them as a negative penalty term to avoid overoptimization in Example 1. We did not use a pure RKHS model with a fixed feature as our reward model. This approach is used by many famous RL papers, such as Yu et al. 19.
>
> 4. While we acknowledge that the generalization bound in Theorem 2, derived under the RKHS assumption, might not be tight, **Our goal is to propose an algorithm that offers both superior experimental performances and fundamental guarantees that formalize the algorithm’s intuition.** We believe that our upper bound suffices for this purpose. Specifically, our intuitive message is that fine-tuned generative models outperform the best designs in offline data by leveraging the extrapolation capabilities of reward models while avoiding the generation of invalid designs. Corollary 1 formalizes this intuition, as mentioned in Section 6.
>
> 5. **Statistical guarantees in most of the literature are not completely tight**, as proving the lower bound is extremely challenging in many scenarios (Wainwright, 19). Therefore, the claim mentioned by the reviewer applies not only to our work but also to nearly all upper bounds in the literature in RKHS/GPs. We think showing tightness is beyond our scope.
>
> **Weakness: the computational cost for the bootstrap method is high since one has to train the model from scratch multiple times. The reward models used in the experiments are mainly single-layer MLPs, and it is doubtful whether this approach is useful when the reward model needs to be a larger model.**
>
> We acknowledge the reviewer’s point; however, we would like to emphasize that (1) computational efficiency is not the primary focus of our paper, (2) it can be addressed straightforwardly by using variants if we want to further accelerate bootstrap part, and (3) the computational cost of bootstrap is not necessarily high in many real applications.
>
> 1. **In scenarios with limited offline data, sample efficiency (the cost of obtaining data with reward feedback) is more critical than computational efficiency.** Our work always prioritizes the former, which differs from online settings where computational efficiency may be more relevant. We have motivated such emphasis in the Introduction by clarifying the hardness of wet lab data collection in scientific domains.
>
> 2. **To increase speed, we can utilize various bootstrap variants (Lakshminarayanan et al. Al, Osband et al., Chua et al.) without training from scratch, such as sharing backbone models, Bayesian deep learning way, etc.** In this way, the bootstrap has been practically and widely used in the ML community. We will add such an explanation.
>
> 3. The computational cost of running pure bootstrap in our experiments is also not so high in many real applications. Indeed, while the reward models in Section 7.1 are not single-layer MLPs (they include transformers and are representative models in genomics proposed in Avsec et.al 21), the training takes several hours. In scenarios where sample efficiency is crucial, training multiple times (or in a parallel manner) won’t be a significant concern. In fact, for both tasks we only employ 3,4 bootstrapped models, so it does not bring too much computational concerns.
>
> **It is unclear whether it is useful given the current experimental results. On one hand, KL regularization is a widely-known technique for preventing over-optimization in diffusion models so it is certain that the KL regularization term will help. On the other hand, the proposed algorithm mixes both the conservative reward estimator and the KL regularization term together, making it unclear which part is playing the role in avoiding over-optimization. My guess is that, for the most part, only the KL regularization term is effective in the end.**
>
> We believe there may be a misunderstanding regarding the interpretation of our experimental results. **In our experiments, we have compared our proposal with the baseline that has the KL regularization, as noted in Section 7 ( Lines 177-185 in our paper), but do not have conservatism in the reward model.** For example, in images, STRL and our proposals BRAID-Boot, BRAID-Bonus) share exactly the same KL strength, and its value ($\alpha=1$) can be found in Table 5, Appendix  F.2.4. Therefore, our current experiments have directly addressed your concern by showing the effectiveness of conservative reward modeling.
>
> **Q: Can be adapted to ODE samplers?**
>
> We address this in the global rebuttal.

---

> > ### Comment · Reviewer_dLuW · 2024-08-07
> > **Thanks for your rebuttal**
> >
> > After careful consideration, I decided to raise my score to 5.

---

### Official Review · Reviewer_wUpc · 2024-07-12

**Soundness:** 3
**Presentation:** 2
**Contribution:** 3
**Rating:** 7
**Confidence:** 3

**Summary:**

1) This paper analyzed the two mainstream angles of computational design.
2) Proposed a hybrid one that offline fine-tunes generative models.
3) Conduct experiments on two tasks to show the performance of their method.

**Strengths:**

1) Sufficient theoretical analysis and detailed preliminaries.
2) The idea is straightforward.
3) The method is comprehensive.

**Weaknesses:**

1) In the introduction, the advantages and disadvantages of the two mainstream issues are not fully analyzed.
2) Insufficient metrics evaluation for image generation task.

**Questions:**

N/A

**Limitations:**

1) Lack of gradual guidance from analyzing the advantages and disadvantages of existing methods to proposing the hybrid method.
2) Lack of multi-metric quantitative results analysis. (e.g. in the image generation task, the fidelity and diversity should also be reported by like LPIPS Score/CLIP score/FID/IS, etc.)

---

> ### Author Rebuttal · Authors · 2024-08-06
>
> Thank you for your positive feedback! We have addressed your concerns by providing additional explanations on (1) the disadvantages and advantages of pure generative model/MBO approaches, and (2) experimental results on diversity metrics based on CLIP scores.
>
> **Weaknesses: In the introduction, the advantages and disadvantages of the two mainstream issues are not fully analyzed. Limitations: Lack of gradual guidance from analyzing the advantages and disadvantages of existing methods to proposing the hybrid method.**
>
> We have tried explaining well; however, due to space constraints, it may come across as somewhat terse. Assuming we have more space in the final version, we will elaborate further in the introduction as follows.
>
> 1. Pure generative approach: We primarily refer to conditional diffusion models.
> - Disadvantages: It may be unclear whether we can learn designs with properties superior to those in the offline data. We will explicitly connect the introduction with Section 2. This connection will clarify that we indeed compare our approach with conditional diffusion models as a pure generative approach in Section 7.
> - Advantages: We can generate valid designs that lie within the sample space (e.g., natural image space, natural DNA, RNA, molecular space), which is challenging to achieve with a pure MBO approach.
>
> 2. Pure MBO approach
> - Disadvantages: We will add the following example in the introduction. It can be challenging to incorporate the natural constraints of the valid space. For example, when the design space is an image space and the reward function is compressibility, optimizing the just learned reward model may result in highly non-natural images with high compressibility. However, our typical goal is to obtain a natural image with high compressibility. To achieve this, it is essential to incorporate a (pre-trained) generative model that characterizes the natural image space.
> - Advantages: Our approach learns a reward model, granting us the extrpolation capabilities so that we can generate designs beyond the offline data distribution by leveraging the extrapolation capabilities of the reward model.
>
> **Weakness: Lack of multi-metric quantitative results analysis. (e.g. in the image generation task, the fidelity and diversity should also be reported by like LPIPS Score/CLIP score/FID/IS, etc.)**
>
> Thank you for the suggestion. As the primary goal of the experiments is to demonstrate that our method can mitigate overoptimization, we focus on reporting the performance metrics needed to show this in Section 7.2. However, we recognize that general synthesizing metrics are frequently reported in papers on diffusion models (though they may not be directly helpful in measuring overoptimization). The following is our attempt at a similarity score.
>
> **Our attempt**: While achieving diversity is not our stated promise, we experimented with CLIP cosine similarities for completeness to measure the diversity of generated samples. However, we observed that CLIP similarities are influenced largely by semantic information. For example, two images generated with the same prompt (e.g., cat) are likely to have a high CLIP cosine similarity (>0.8), even if they are visually very different. We agree that LPIPS might be more reliable.

---

### Author Rebuttal · Authors · 2024-08-06

We appreciate feedback from all reviewers. We respond to raised weaknesses/questions as much as possible.

**Papers we cite:** We have added the papers we cited in our response here.

* Deringer, V. L., Bartók, A. P., Bernstein, N., Wilkins, D. M., Ceriotti, M., & Csányi, G. (2021). Gaussian process regression for materials and molecules. Chemical Reviews, 121(16), 10073-10141.

* Gramacy, Robert B. Surrogates: Gaussian process modeling, design, and optimization for the applied sciences. Chapman and Hall/CRC, 2020.

* Taskiran, Ibrahim I., et al. "Cell-type-directed design of synthetic enhancers." Nature 626.7997 (2024): 212-220.

* Yu, Tianhe, et al. "Mopo: Model-based offline policy optimization." Advances in Neural Information Processing Systems 33 (2020): 14129-14142.

* Wainwright, Martin J. High-dimensional statistics: A non-asymptotic viewpoint. Vol. 48. Cambridge university press, 2019.

* Chen, Ricky TQ, et al. "Neural ordinary differential equations." Advances in neural information processing systems 31 (2018).

* Lakshminarayanan, Balaji, Alexander Pritzel, and Charles Blundell. "Simple and scalable predictive uncertainty estimation using deep ensembles." Advances in neural information processing systems 30 (2017).

* Chua, Kurtland, et al. "Deep reinforcement learning in a handful of trials using probabilistic dynamics models." Advances in neural information processing systems 31 (2018).

* Osband, Ian, et al. "Deep exploration via randomized value functions." Journal of Machine Learning Research 20.124 (2019): 1-62.

* Rombach, Robin, et al. "High-resolution image synthesis with latent diffusion models." Proceedings of the IEEE/CVF conference on computer vision and pattern recognition. 2022.

* Avdeyev, Pavel, et al. "Dirichlet diffusion score model for biological sequence generation." International Conference on Machine Learning. PMLR, 2023.

* Xu, Minkai, et al. "Geodiff: A geometric diffusion model for molecular conformation generation." International Conference on Learning Representations. 2022

**Additional rebuttal for a reviewer dLuW**

That’s an interesting point. Here are several ideas. We will incorporate the following into the main text:

- While somewhat heuristic, we can still use the ODE sampler after fine-tuning. Specifically, when generating samples for fine-tuning, we use the SDE sampler with a small variance term; however, after fine-tuning, we can still use it as the ODE sampler by setting the variance term to 0.

- Another idea is to explicitly calculate (or estimate) the log-likelihood ($log p_{\text{pre}}$) using the well-known formula from the neural ODE paper (Theorem 1 in Chen et.al) and incorporate it as a penalty in the reward term. However, the caveat of this approach is that estimating this log-likelihood is often challenging and non-differentiable.

---

### Decision · Program_Chairs · 2024-09-25

**Decision:**

Accept (poster)

**Comment:**

The paper proposes a conservative fine-tuning approach to tackle overoptimization in diffusion models, which is common in settings where task-specific reward models are unknown and offline data is used instead for fine-tuning diffusion models. The reviewers highlighted the originality of the proposed method and found that the empirical evaluation satisfactorily demonstrated the method's usefulness. All reviewers agree that the paper should be presented at NeurIPS, and I agree that the paper would be of interest to the NeurIPS community and would be a valuable addition to the program.